# AnoShift: A Distribution Shift Benchmark for Unsupervised Anomaly Detection

Marius Dragoi*[1], Elena Burceanu*[1,2], Emanuela Haller*[1,3], Andrei Manolache[1], and Florin Brad[1]

[1]Bitdefender, Romania
[2]University of Bucharest
[3]Politehnica University of Bucharest

{mdragoi,eburceanu,ehaller,amanolache,fbrad}@bitdefender.com

## Abstract

Analyzing the distribution shift of data is a growing research direction in nowadays Machine Learning (ML), leading to emerging new benchmarks that focus on providing a suitable scenario for studying the generalization properties of ML models. The existing benchmarks are focused on supervised learning, and to the best of our knowledge, there is none for unsupervised learning. Therefore, we introduce an unsupervised anomaly detection benchmark with data that shifts over time, built over Kyoto-2006+, a traffic dataset for network intrusion detection. This type of data meets the premise of shifting the input distribution: it covers a large time span (10 years), with naturally occurring changes over time (*e.g.* users modifying their behavior patterns, and software updates). We first highlight the non-stationary nature of the data, using a basic per-feature analysis, t-SNE, and an Optimal Transport approach for measuring the overall distribution distances between years. Next, we propose **AnoShift**, a protocol splitting the data in IID, NEAR, and FAR testing splits. We validate the performance degradation over time with diverse models, ranging from classical approaches to deep learning. Finally, we show that by acknowledging the distribution shift problem and properly addressing it, the performance can be improved compared to the classical training which assumes independent and identically distributed data (on average, by up to 3% for our approach). Dataset and code are available at `https://github.com/bit-ml/AnoShift/`.

## 1 Introduction

Analyzing and developing Machine Learning algorithms under gradual distribution shifts is a problem of high interest in the research community. There is a growing enthusiasm for building benchmarks over existing or new datasets [26, 22, 44, 6, 20], that formulate a setup for isolating the shifting aspect and create a better ground for this research field. A better understanding of the distribution shift problem might lead to findings of underlying fundamental aspects, shedding new light on robustness and generalization problems. We argue that the distribution shift occurs naturally and gradually in a continuous data stream (*e.g.* monitoring network traffic), allowing an in-depth analysis of the problem. On the other side, artificially generated scenarios usually exhibit sudden changes that do not simulate the natural shift problem. Yet, the annotation process for streaming data is quite difficult and expensive, considering the massive amount of data.

---

*Equal contribution. [1]Bitdefender Theoretical Research Team: `https://bit-ml.github.io/`

36th Conference on Neural Information Processing Systems (NeurIPS 2022) Track on Datasets and Benchmarks.

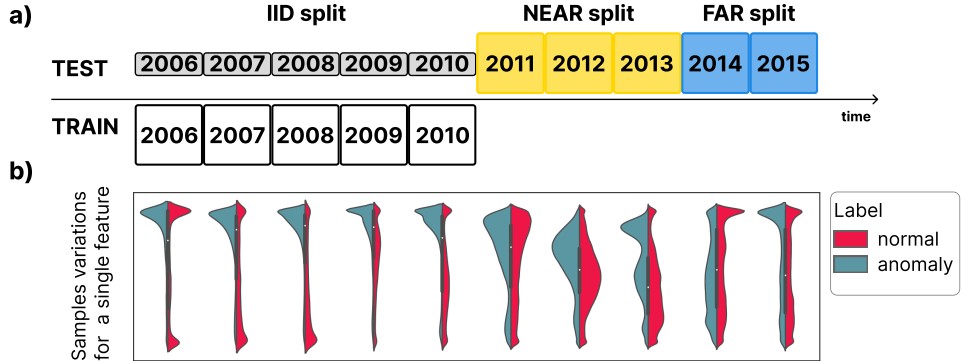

Figure 1: **a)** The proposed **AnoShift** splits over Kyoto-2006+ dataset. The `IID` (gray) testing split comes from the same temporal span as the `TRAIN` set (white), while `NEAR` (yellow) and `FAR` (blue) splits are from different time spans, with `NEAR` being closer to the training set than `FAR`. **b)** To highlight the utility of the proposed chronological protocol, we exemplify the continuous evolution of data, illustrating the distributions of normal and anomaly samples over the considered 10 years. We exemplify the evolution of the percent of recent connections that have the same source and destination IP addresses as the current connection (feature 9 - Dst host srv count).

From a practical point of view, continuous IT infrastructure monitoring has become essential for computer security and resilience. Recent anomaly detection and intrusion detection systems (IDS) obtain strong results on specific datasets but drastically fail in real-world scenarios [47]. Our experimental analysis proved a natural change of the Kyoto-2006+ data over the 10 years period when the data was collected. The shift is noticeable both over the input distribution and considering the performance of several anomaly detection systems. Several reasons behind the observed shift are: users leaving or coming to the network, per user interest changes leading to network interaction changes, updates to the software versions, patching old vulnerabilities but revealing new attack vectors for intruders.

To better assess the models' capabilities, we introduce a chronology based evaluation protocol, distinctly evaluating performance on test data splits (`IID`, `NEAR` and `FAR` - Fig. 1) with different temporal distances towards the training set (`TRAIN` -Fig. 1). We observe that the performance of anomaly detection models consistently degrades when tested on data from longer time horizons. Moreover, we prove that a basic distillation technique overcomes a classic IID (assuming independent and identically distributed data) training under gradual data shifts, proving that the awareness of the shift problem might lead to better solving the task.

Summarized, our **main contributions** are the following:

- We analyzed a large and commonly used dataset for the unsupervised anomaly detection task in network traffic (Kyoto-2006+) and demonstrated that it is affected by distribution shifts. The per-feature distributions and t-SNE show multiple changes over the years, and the Optimal Transport Dataset Distance gave us an estimate of its magnitude.

- We propose a chronology-based benchmark, which focuses on splitting the test data based on its temporal distance to the training set, introducing three testing splits: `IID`, `NEAR`, `FAR` (Fig. 1). This testing scenario proves to capture the in-time performance degradation of anomaly detection methods for classical to masked language models. This benchmark aims to enable a better estimate of the model's performance, closer to the real world performance.

- We prove that properly acknowledging the distribution shift may lead to better performing anomaly detection models than classical IID training. When facing distribution shift, a basic distillation technique positively impacts the performance by up to 3% on average.

## 2 Related work

**Relation to benchmarks targeting distribution shift** Recently, there has been an increased amount of effort and focus in this direction, with several benchmarks emerging. They emphasize the non-

stationary nature of the data, with various underlying reasoning. The most common approach is to search for gaps in the input data distribution that appear with time [26, 22], taking into perspective that the world is continuously evolving; therefore, the data acquired continuously from it should exhibit the same behavior. Our work aligns with this perspective by working with traffic logs from a large university network over 10 years. In [26], the authors focus on how the appearance of basic objects changes from year to year, while [22] emphasizes the seasonal patterns that appear in news language (*e.g.* elections, hurricanes). A second axis exploited for noticing shifts in data is the spatial one. In [6], geolocalization is used in conjunction with the time for guiding the shift. In [20], the gap is based on higher level characteristics, like x-ray data from different hospitals, but also on geolocalization. In searching for the autonomous driving robustness, a more complex variation is provided in [44] following the weather, time of day, and congestion levels. Nevertheless, all works analyze the distribution shift for supervised tasks, focusing on NLP or Computer Vision. In [22], the authors monitor the evolution of the perplexity metric, with models learned in a self-supervised manner as a masked language model. They emphasize the need to link the shift analysis to a downstream task, several supervised ones in their case. Differently, **AnoShift**, our benchmark proposal, tackles an unsupervised anomaly detection task under non-stationary data.

**Relation to traffic anomalies**    Models tackling Network Intrusion Detection are covered by lots of surveys [16, 2, 19], structured around dataset variations, anomaly types, and methods variation. A fair amount of the approaches are supervised [34], based on tree classifiers [48], modeling the task as a binary or multi-class anomaly (intrusion) classification. But we are interested here in the unsupervised setup [31]. Usually, the best models are quite simple, most of them are shallow [17], based on OC-SVM [39] or Isolation Forest [27], or very small neural nets [31]. Several solutions introduce deep learning approaches for intrusion detection [33], transforming the data into images [13], or modeling the problem using GNNs [29].

An important problem we identified in this area is that the datasets used for the task are easily saturated, mainly because they either lack variety (*e.g.* simulated traffic patterns for anomalies) or have a very few annotated anomalies, or are small-scale, covering only several days [12, 45, 40, 37, 38, 32, 18, 7, 34]. In contrast, Kyoto-2006+ [43] spans over 10 years (2006-2016), containing continuous natural traffic logs from a large university network, within a sub-net of honeypots. Most of those datasets cover basic networks, but there are some oriented towards IOT traffic [38], or even to the autonomous driving field, Internet of Vehicles [48]. But another reason for saturation, is the IID training setup, as we will show in this work. These generalization problems are very acute, leading to weak performances for those algorithms when applied on real world data, or on a new dataset [47]. With **AnoShift**, we highlight the IID training problem, by proposing a different training and evaluation setup based on temporal distances, closer to a realistic case.

# 3    Chronological protocol

We introduce a chronological protocol for building train and testing splits that can highlight the temporal evolution of data. Taking into consideration the timestamps of our data, we propose to build a training split (TRAIN) along with three different testing splits (IID, NEAR and FAR), comprising multiple years of data (Fig. 1a)). The TRAIN and IID splits are extracted from the first period of time, and the IID tests should highlight the expected performance when there is no distribution shift between train and test. The NEAR and FAR splits are each extracted from different periods of time, where NEAR is closer to the training data and FAR is farther away. We expect standard models to exhibit better performance on NEAR compared to FAR, which we experimentally prove in Sec. 4.2. Our proposed benchmark will provide a better estimate of the expected performance when the model is deployed in the wild and exposed to the inevitable distribution shift of the data. To the best of our knowledge, **AnoShift** is the first to provide a proper scenario for studying the generalization capabilities of unsupervised learning models for anomaly detection.

Our work revolves around Network Intrusion Detection Systems (NIDS), tackling the problem of distribution shifts that naturally appear in network traffic data. We work over the popular Kyoto-2006+ dataset (Sec. 3.1), which was collected over ten years, providing us with enough data to capture the temporal evolution. Starting from Kyoto-2006+, we introduce our **AnoShift** Benchmark (Sec. 3.2) that proposes one training and three testing splits, which highlight the difficulty of dealing with data temporally distant from the training set.

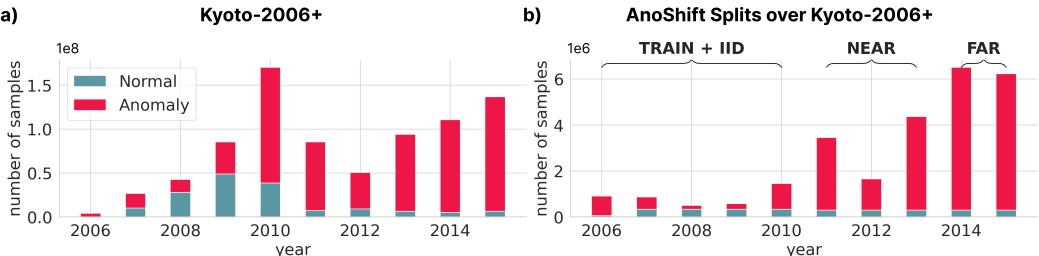

Figure 2: **a)** Yearly splits of the network traffic data from Kyoto-2006+ dataset, highlighting the proportion of normal and anomaly samples. **b)** Proposed train and test splits in our **AnoShift** benchmark. Considering that `TRAIN` and `IID` splits are sampled from the same time span, we have jointly represented them. Note that while for `TRAIN`, `NEAR` and `FAR` we extract the same number of normal samples per year, the `IID` split contains 10 times less normal samples. The anomaly samples are extracted such that we maintain the normal vs. anomaly proportion of the original data.

## 3.1 Kyoto-2006+

Kyoto-2006+ [43] is a reference dataset for Anomaly Detection over network traffic data [35]. It is built on 10 years of real traffic data (Nov. 2006 - Dec. 2015), captured by a system of $348$ honeypots in $5$ sub-networks inside the Kyoto University. Briefly, a honeypot is a real or virtual machine simulating a regular computer (having an OS and multiple services running on it). Its purpose is to deceive an attacker into taking advantage of the vulnerabilities present on the honeypot machine (*e.g.* software not updated). A honeypot does not request any connection on its own. So in such a scenario, almost all traffic coming to a honeypot machine is unsolicited and therefore considered malicious. By design, this type of dataset has a large percent of anomalies (89.5% anomalies in Kyoto-2006+) compared to other anomaly detection datasets. The 14 conventional features of the dataset include 2 categorical ones like connection service type or flag of the connection and 12 numerical like the connection duration or the number of source bytes. We put more details about Kyoto-2006+ in Appendix A. This dataset is spread across a very large period of time, and it contains exclusively real-world traffic, without simulated events.

## 3.2 AnoShift benchmark

To keep the natural distribution shift of the network traffic data, we sample a fixed number of normal samples per year (#months×25k for `TRAIN`, `NEAR`, and `FAR` and #months×2.5k for `IID`). The number of anomalous samples is chosen such that we maintain the proportion of normal vs. anomaly samples from the original yearly subset. We illustrate this process in Fig. 2. In Fig. 1 b) we illustrate the continuous evolution of the data features over the considered 10 years, comparing the distribution for one feature (feature 9 - Dst host srv count). Such behavior can be observed for the majority of features, a fact highlighted by our in-depth analysis from Sec. 4.1. The `TRAIN` and `IID` samples are collected from $[2006-2010]$, while the `NEAR` and `FAR` splits consist of $[2011-2013]$ and $[2014-2015]$ intervals. The protocol is illustrated in Fig. 1 and Fig. 2.

### 3.2.1 Experimental setup

**Preprocess network traffic data** We use the 14 conventional features from the new version of the Kyoto dataset [2006-2015] and convert 3 of the 12 numerical features to categorical values by using an exponentially-scaled binning method between 0 and the maximum value of each feature, such that the bins have a higher density for smaller values and get increasingly wider towards larger values. We used a basis of 1.1, which results in 233 bins, where the width of the $i$th bin is given by: $bin_i = [1.1^i - 1, 1.1^{i+1} - 1]$. We keep the original percentage features (9 out of 12 numerical features), which are discretized in 100 values. Therefore, our preprocessing results in a fixed vocabulary size and each possible token is known apriori. See in Fig. 3 a preprocessed sample. Our processing of the original dataset does not pose any privacy concerns since it does not contain any sensitive information, such as IP address. However, data binning constitutes another potential limitation in our work.

| | Duration | Service | Source bytes | Destination bytes | Count | Same srv rate | Serror rate | Srv serror rate | Dst host count | Dst host srv count | Dst host same src port rate | Dst host serror rate | Dst host srv serror rate | Flag | Label |
|---|---|---|---|---|---|---|---|---|---|---|---|---|---|---|---|
| 76349 | c020 | http | c252 | c360 | 16 | 0.19 | 0.0 | 0.0 | 1 | 1 | 0.0 | 0.0 | 0.0 | RSTO | 1 |
| 35517 | c043 | ssh | c277 | c379 | 0 | 0.00 | 0.0 | 0.0 | 0 | 0 | 0.0 | 0.0 | 0.0 | S1 | 1 |
| 42930 | c00 | dns | c20 | c30 | 0 | 0.00 | 0.0 | 0.0 | 1 | 1 | 0.0 | 0.0 | 0.0 | SHR | 1 |
| 10277 | c016 | smtp | c282 | c357 | 1 | 1.00 | 0.0 | 0.0 | 1 | 100 | 0.0 | 0.0 | 0.0 | SF | 0 |
| 32392 | c045 | ssl | c2115 | c392 | 0 | 0.00 | 0.0 | 0.0 | 1 | 1 | 0.0 | 0.0 | 0.0 | RSTR | 1 |

Figure 3: Examples of preprocessed Kyoto-2006+ instances. See Appendix A for details.

**Metrics for anomaly detection**     To analyze the performance of various models on our proposed benchmark, we use the labels (normal and anomaly) provided by the Kyoto-2006+ dataset. As we deal with imbalanced sets, we study the ROC-AUC metric and also evaluate the PR-AUC metric, for both inliers and outliers (note that for a random classifier, PR-AUC for a specific class is close to the ratio of data in that specific class). We report the IID, NEAR and FAR performances as the arithmetic mean of performances over their associated yearly splits.

# 4    Distribution shift analysis

We perform an in-depth analysis of the proposed benchmark from three points of view. **First**, we study the inherent non-stationarity of the considered data, highlighting the natural shift between the years, considering both simple, per feature metrics and more complex metrics between distributions (Sec. 4.1). **Second**, we analyze various anomaly detection models, highlighting the performance decrease when dealing with testing data that is temporally distant from the training set (Sec. 4.2). **Third**, we discuss the importance of acknowledging the data shift and emphasize the positive impact of a basic distillation technique over the standard IID approach (Sec. 4.3). We add supplementary discussions on the method in Appendix A.1.

We run our experiments on an internal cluster with multiple GPU types: GTX 1080 Ti, GTX Titan X, RTX 2080 Ti, RTX Titan. We estimate that we need 5 days to reproduce the experiments on 1 GPU. The CPU training for OC-SVMs, IsolationForest, and LOF benchmarks takes  3 days.

## 4.1    Inherent non-stationarity

**Visualization of the data shifts**     For a visual interpretation of the yearly shift, we have considered the unsupervised t-SNE [46] to illustrate the high dimensional data structure (PCA visualization available in Appendix  A). In Fig. 4 we introduce the comparison between pairs of yearly splits and the whole figure can be interpreted as a similarity matrix, each cell $(i, j)$ illustrating the similarity between point clouds of year $i$ vs. year $j$. Each row illustrates the point clouds of the corresponding year over all the other point clouds. At the same time, each column presents the point clouds of the corresponding year below all the other point clouds for a better understanding of the distribution shifts. We observe that point clouds move away as we increase the temporal gap between their corresponding years. This confirms our intuition that the analyzed network traffic data is continuously shifting in time and emphasizes the need for a benchmark as **AnoShift** that can efficiently test the robustness of models under this inherent non-stationarity of natural data.

**Per-feature shift**     We further analyze whether the dataset's statistics at the feature level are changing from one year to another. Recall that we have 2 categorical features and 12 numerical ones. We extract the normalized histogram per year for each feature and compute the Jeffreys divergence [15] between those histograms. The Jeffreys divergence is a commonly used symmetrization for Kullback-Leibler divergence [21]: $KL(p, q) + KL(q, p)$, and it is proven to be both symmetric and non-negative. We highlight that such an analysis can only illustrate simple scenarios, studying the distribution change from the perspective of single feature changes. With all the considered baselines from Sec.4.2, we have observed a significant decrease in performance for the years 2014 and 2015, leading to the intuition that this subset may have substantial differences from the others. Consequently, in Fig. 5, we illustrate the Jeffreys divergence for two features that we find to have a large 2014-2015 distance, but also for a third one that has significant high values in the distance map on other years than the two.

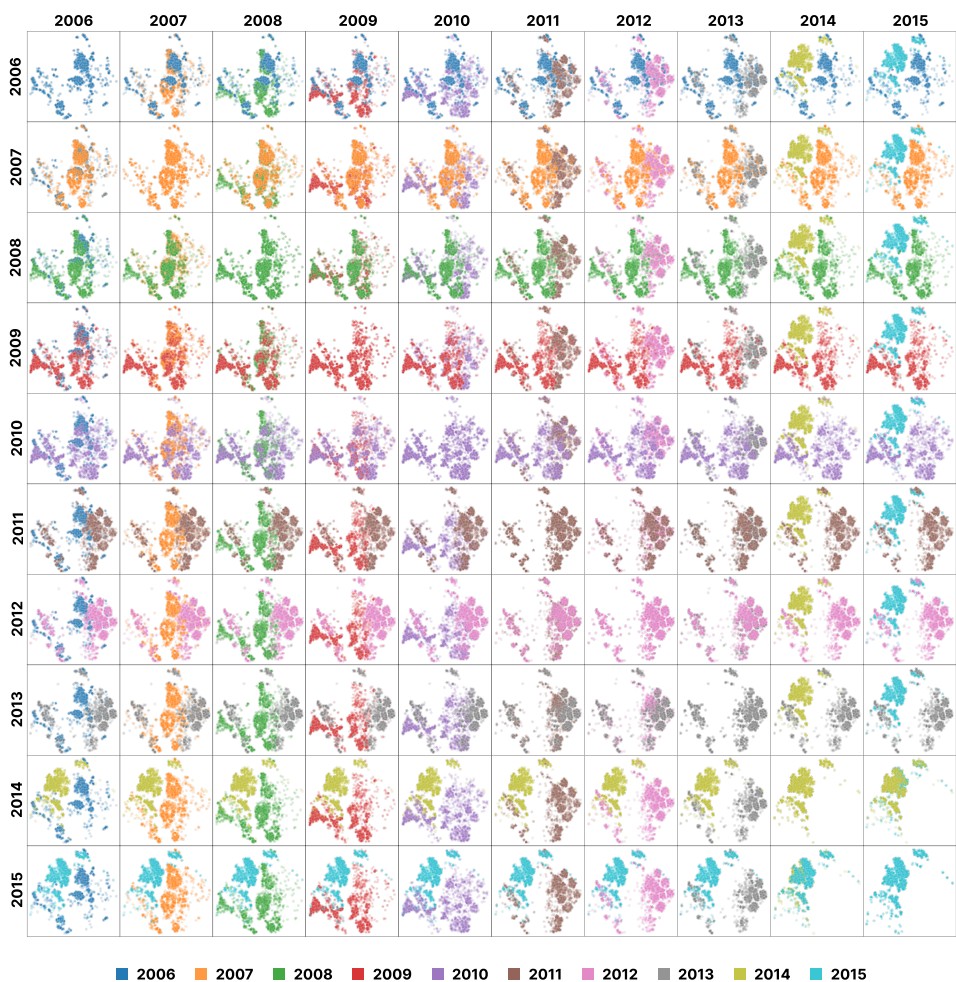

Figure 4: Comparison between yearly splits using t-SNE visualization. We observe that the discrepancy between point clouds increases with the temporal distance between splits, colors becoming more separated over time. The analysis is performed considering 2k randomly sampled points per split. Follow the 2007 row: see the orange cluster on top of clusters associated to the other years. It is very similar to its neighbours 2006-2008, and the similarity diminishes in time (see 2015).

**General shift** We next explore the distribution differences between dataset splits over time by using the Optimal Transport Dataset Distance method (OTDD) [3]. OTDD relies on optimal transport, a geometric method for computing distances between probability distributions for comparing datasets. This analysis shows how the splits move away from each other over time (see Fig. 6). Compared with the per feature approach, this method allows us to gain a better intuition for the performance on a new split, giving us a single distance based on all features. We observe how the inliers (first image) nicely distances in OTDD value, directly correlated with the distance in time. As for the outliers (third image), it is noticeable that they are quite different between the splits of the first years. We notice that the distances between inliers and outliers (in the middle) show that FAR years' outliers are similar to TRAIN years' inliers, an observation that we empirically confirm in Tab. 1, where all models suffer from a steep descent in performance (bellow random). We run the method with the default parameters for DatasetDistance, over the standardized input of Kyoto, with one-hot encoded categorical variables, 3 times, with a randomly sampled 5k entries per year.

## 4.2 Impact on IID models

We introduce the **AnoShift** benchmark to understand better the impact of data shifts that naturally appear over time on the performance of anomaly detection models. We hope that the proposed splits

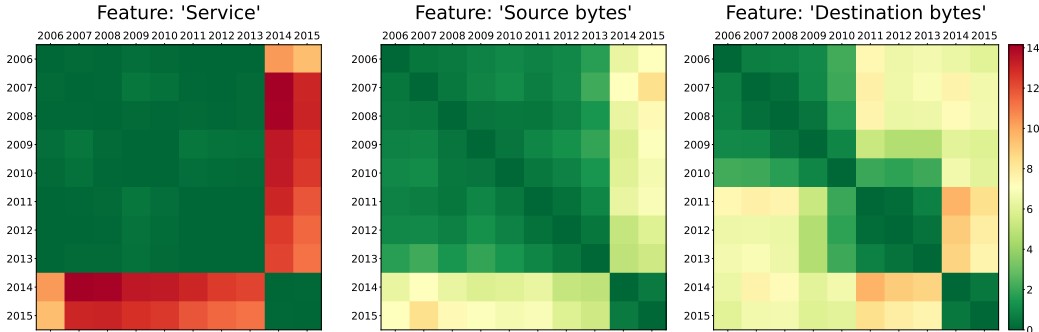

Figure 5: Jeffreys divergence between Kyoto years. First two images represent features with a large 2014-2015 distance. The $3^{rd}$ one is for a feature with significant difference between the histograms across years. Note that it is difficult to predict the performance of the method on a new split, only based on those per feature distances between distributions.

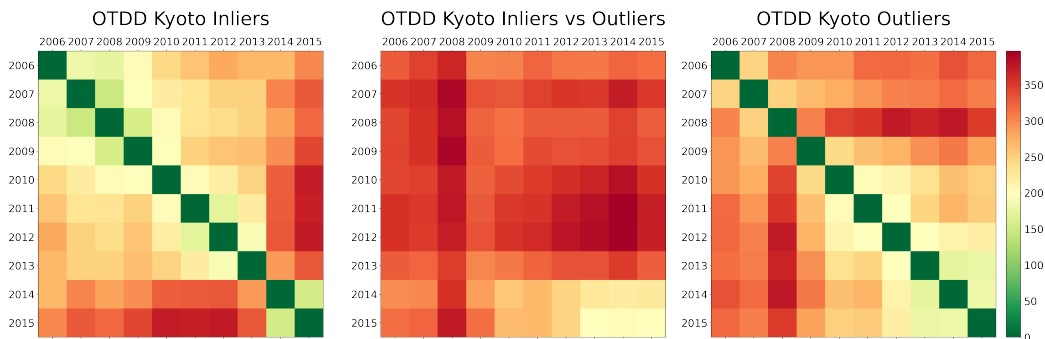

Figure 6: Optimal Transport Dataset Distance for Kyoto. See distances between inliers (first), inliers and outliers (second), and outliers (third). The distances from inliers generally increase as you move further from the diagonal, showing large distances between TRAIN and FAR data. Moreover, notice in second image how outliers in the FAR splits are quite similar with inliers from TRAIN, also explaining the abrupt performance drop on farther data (Tab. 1).

will push forward the research in this direction and help build more robust models that can deal with mild to severe distribution changes between test and training sets. In this context, in the current section, we will study the performance degradation of various anomaly detection approaches, from IID to NEAR and FAR testing splits.

**Anomaly detection models**  We have considered several unsupervised baselines, ranging from more classical approaches, like Isolation Forest [27], OC-SVM [39], LocalOutlierFactor(LOF) [5] and recent ECOD [24] and COPOD [23], to deep learning ones, like SO-GAAL [28], deepSVDD [36], AE [1] for anomalies, LUNAR [14], InternalConstrastiveLearning [41] and our proposed transformer for anomalies model, based on the BERT [11] architecture. For part of the baselines, we have employed the PyOD library [49].

**BERT for anomalies**  We use a simplified BERT architecture, without pretraining, with around 340k trainable parameters. We train the BERT model as a Masked Language Model (MLM), using a data collator that randomly masks a fraction $p$ of the input sequence and optimizing a cross-entropy loss function between the model predictions at mask positions and the original tokens. We derive a sequence anomaly score by randomly masking a fraction $p$ of tokens in the sequence and averaging the probabilities of the correct tokens at mask positions given by the classification layer over the vocabulary. At evaluation time, we average the score over 10 mask samplings. A detailed description of the model is introduced in Appendix A. In our experiments, we used $p = 15\%$.

Table 1: Performance evolution over time, for classical and deep methods: `IID` vs `NEAR` vs `FAR`. Notice that the ROC-AUC is dropping over time in all cases, except for BERT and SO-GAAL methods, showing this is a property of the method, rather than a problem with the dataset. More precisely, those methods model the outliers very well in the `NEAR` split (see PR-AUC-out), while the PR-AUC-in is dropping, confirming the distribution shift over time (see Appendix A). The variance for `FAR` is the highest and almost all methods perform under-random on it. Best scores per split are shown in bold: `NEAR`-best is BERT, but interestingly, `IID`-best is LOF, and `FAR`-best is COPOD. PR-AUC for inliers and outliers are available in Appendix A-Fig. 10 and in Tab. 2.

| | | ROC-AUC ↑ | | |
|---|---|---|---|---|
| Type | Baselines | IID | NEAR | FAR |
| Classical | **OC-SVM** [39] (train 5%) | $76.86 \pm 0.06$ | $71.43 \pm 0.29$ | $49.57 \pm 0.09$ |
| | **IsoForest** [27] | $86.09 \pm 0.54$ | $75.26 \pm 4.66$ | $27.16 \pm 1.69$ |
| | **ECOD** [24] | $84.76$ | $44.87$ | $49.19$ |
| | **COPOD** [23] | $85.62$ | $54.24$ | $\mathbf{50.42}$ |
| | **LOF** [5] | $\mathbf{91.50} \pm 0.88$ | $79.29 \pm 3.33$ | $34.96 \pm 0.14$ |
| Deep | **SO-GAAL** [28] | $50.48 \pm 1.13$ | $54.55 \pm 3.92$ | $49.35 \pm 0.51$ |
| | **deepSVDD** [36] | $73.43 \pm 0.94$ | $69.61 \pm 0.83$ | $31.81 \pm 4.54$ |
| | **AE** [1] **for anomalies** | $81.00 \pm 0.22$ | $44.06 \pm 0.57$ | $19.96 \pm 0.21$ |
| | **LUNAR** [14] (train 5%) | $85.75 \pm 1.95$ | $49.03 \pm 2.57$ | $28.19 \pm 0.9$ |
| | **InternalContrastiveLearning** [41] | $84.86 \pm 2.14$ | $52.26 \pm 1.18$ | $22.45 \pm 0.52$ |
| | **BERT [11] for anomalies** | $84.54 \pm 0.07$ | $\mathbf{86.05} \pm 0.25$ | $28.15 \pm 0.06$ |

In Table 1 we report the results of our experiments. Each baseline model was trained 3 times with a basic set of hyperparameters, and we reported the average results and the standard deviation. Both the OC-SVM and the LUNAR model were trained solely on $5\%$ of the `TRAIN` set to reduce the computational burden. For all of the considered models, except ECOD, we observe a performance degradation between `NEAR` and `FAR` splits, highlighting that these anomaly detection models cannot cope with the distribution shift. In the case of ECOD, the performances of both `NEAR` and `FAR` splits are below random, making their relative order irrelevant. The `IID` evaluation, which is the most popular methodology, proves to give an illusion of high performance, as the performance quickly degrades once we consider a testing set from a different period. The evolution is also presented in Appendix A-Fig. 10, illustrating ROC-AUC along with PR-AUC for inliers and outliers. We observe a rapid degradation for inliers PR-AUC, indicating that normal data distribution is continuously changing, and the outliers detection may not be reliable. These experiments highlight the issues of current anomaly detection models and prove the benefits of the **AnoShift** benchmark.

**Performance on `FAR`**    With all tested baselines, we notice a significant decrease in performance for 2014-2015 years for inliers, which motivates us to further investigate the particularities of this subset. We observe a large distance in the Jeffreys divergence between 2014-2015 and the rest of the years for 2 features: service type and the number of bytes sent by the source IP (see Fig. 5). From the OTDD analysis in Fig. 6, we observe that: first, the inliers from `FAR` are very distanced to training years; and second, the outliers from `FAR` are quite close to the training inliers. One root cause of those events can be the steep increase of the "DNS" traffic percentage (from 4% to 37%, in 2013, and 2014 respectively). This contributes to the distribution shift on `FAR`, explaining the low performance.

**Monthly evaluation**    In Fig. 7, we take a closer look at the BERT's performance at month granularity and break down performance on inliers and outliers. First, notice how the inliers' performance gradually degrades over time, to an abrupt drop at farther months. This doubles the analysis from Sec. 4.1, where we notice the difference between the `TRAIN` years and `FAR` (through Jeffreys and OTDD experiments). Second, we observe that on `IID` years, the anomalies are modeled quite poorly by our language model, resulting in a slightly lower `IID` performance in comparison with `NEAR`.

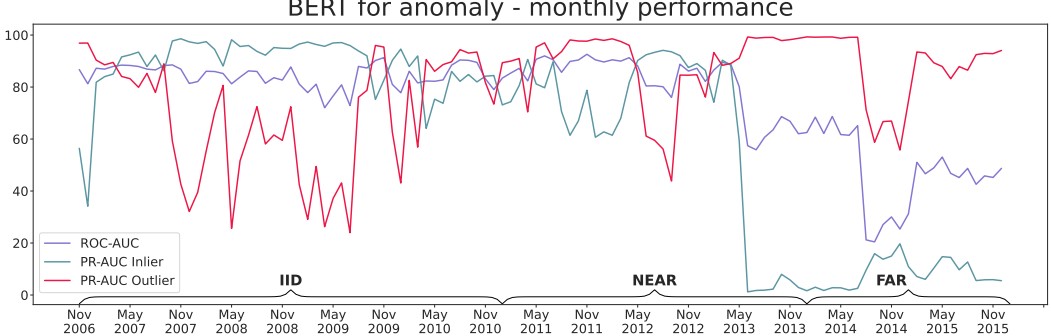

Figure 7: BERT for anomaly, evaluated on each month. We show the ROC-AUC, PR-AUC for inliers, and PR-AUC for outliers. The performance for the inliers is slowly decreasing during `IID` and `NEAR` splits, dropping suddenly just before the `FAR` split, showing how the language model fails to recognize inliers once it moves further apart from the training data. On the other hand, there are parts of the `IID` split where the outliers are quite poorly modeled, explaining the slightly poor performance of BERT on `IID` when compared with `NEAR` split.

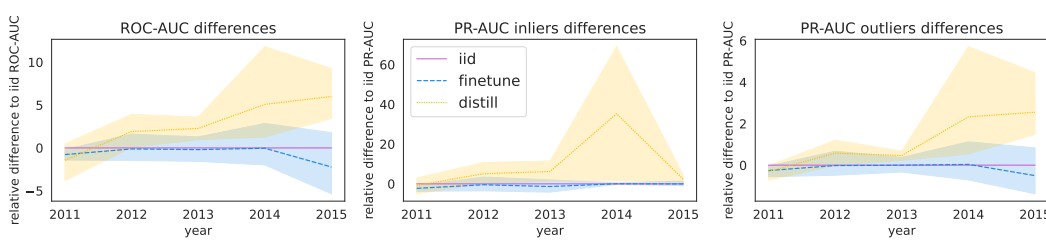

Figure 8: ROC-AUC, PR-AUC-in, PR-AUC-out for Finetune and Distill strategies, relative to the iid. The performance is averaged over all training subsets. Even though the strategies have a high variance in general, the distill is clearly more robust over time when compared to iid and finetune.

## 4.3 Addressing the shifted data

We next compare the performance of a BERT model in 3 training regimes: iid, finetune, and knowledge distillation, for subsets of 300k entries from each year. We use 2006-2010 as training data and evaluate 2011-2015 as individual splits. First, in the **a) iid mode**, we use sets of data starting from 2006 and gradually add each successive split from the train period, initializing a new model for each subset. Next, in the **b) finetune mode**, we start from the iid model trained on 2006 and gradually finetune it on each successive year in the train period. Finally, in the **c) distillation mode**, we start from the iid model of 2006 and reinitialize a same-sized model for each new split, which becomes a student for the previous model by combining the MLM loss with a KL divergence loss with the teacher predictions on the current split. The best performance is achieved by the final distilled model for every test split (see Fig. 8), outperforming iid and finetune by over $3\%$ on average in ROC-AUC. It is worth noting that the effects of distillation are visible over time, with the iid method outperforming it in the first two iterations over the train splits. At all stages, the distillation method obtains the best performance on FAR data, providing a more robust training alternative to distribution shifts in data. The metrics are available in Appendix A-Tab. 3 and pseudocode for the training modes is available in Appendix A.2.

## 4.4 Discussions

**MLM as anomaly detector**   Even though the BERT model greatly exceeds the number of parameters and the complexity of other classical baselines, its generalization performance on farther data is extremely low. The anomaly performance in our case is based on the perplexity score when predicting several masked features in the sample. So if the features are not correlated, the MLM model might be

unable to learn something useful, which might result in learning some specific training set biases, failing to generalize on temporarily distant data (eg. lower score on FAR wrt other baselines). We did not investigate this, but we consider it an interesting direction for future work.

**MLM with the training vocabulary**    In a real world setup, we expect that the fraction of tokens that are previously unseen during training increases with temporal distance. The evaluation score might get artificially inflated due to mapping of unseen features to the UNK token, as for farther points it is easier to predict UNK instead of the right word. Alongside the requirement of a discrete vocabulary, this is another limitation of vocabulary based methods as opposed to other classical approaches. We did not investigate these effects, but it might constitute an interesting direction for future work.

**Other considered datasets**    To emphasize the Kyoto-2006+ value, we briefly discuss here the other considered datasets and why we choose it in the end. We performed an in depth analysis over a large number of datasets, looking for two characteristics, essential for a distribution shift benchmark: it spreads over a large enough time-span, such that the distribution shift will naturally occur, rather than being synthetically injected, exhibiting sudden changes, and it is not solved already (existing methods do not report perfect scores on it). We first looked over a wide range of known **1. network traffic datasets** for intrusion detection, and after analysing them we concluded that most are artificially created, with injected samples, in very restricted scenarios. Only Kyoto-2016 was a proper one, extended over a long enough period of time for showing a natural distribution shift. We next focused our attention on **2. system logs**, since the time-span is usually more extensive in these dataset and the natural distribution shift is more probable to occur. But under our analysis (t-SNE, Jeffreys divergence, OTDD, multiple baselines), these datasets did not exhibit a clear distribution shift over time, so we decided to further analyse them until concludent results. Finally, we looked over **3. general multi-variate timeseries** datasets, but the most popular ones are quite small and almost perfectly solved already. We leave this exact numbers for the considered datasets in the Appendix A.3.

## 5    Conclusion

Our approach highlights the true dimension of distribution shifts that appear in naturally and continuously evolving data streams. We analyze it in Kyoto-2006+ network traffic dataset that spans over 10 years from multiple angles: visually with t-SNE, statistically with histogram distances, and by measuring its magnitude with an Optimal Transport approach. Next, we propose **AnoShift**, a chronology-based benchmark for anomaly detection, to enable the development of models that generalize better and are more robust to shifts in data. Further, we show that by acknowledging the shift and addressing it, the performance can be improved, obtaining a +3% performance boost using a basic distillation technique.

**Acknowledgments**

We thank Razvan Pascanu for guiding us on how to approach the subject and Ioana Pintilie for helping us with baselines for the rebuttal.

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
