# A   Appendix

**Kyoto-2006+**   When an attack occurs, the honeypot saves the network access pattern and other metadata and, at some point, might decide to reboot the system and rewrite back the original configuration. The authors deployed another machine in the network to generate normal traffic data, with a mailing server and a DNS service for a single domain. All traffic data from this server was labeled as clean (the logs also include other protocols for managing the machine over ssh or http and https). The 14 conventional features of the dataset includes 2 categorical features: connection service type and flag of the connection and 12 numerical features: connection duration, number of source and destination bytes, number of connections with corresponding IP addresses in a timeframe of two seconds and the percentage of connections accessing the same service and their rate of "SYN" errors, the prevalence of the connection's source IP address and the requested service in the past 100 connections to the current destination IP (Kyoto features). The malicious traffic is further labeled using three software solutions: an Intrusion Detection Systems at the network level, an Antivirus product, and a shellcodes and exploits detector. In addition to those, the authors labeled other entries based on their prior history of connections from a specific IP and destination port. Additional features include source and destination IP addresses and ports, timestamp and protocol. We note that the generality of the original dataset imposes several limitations for our benchmark. More specific, the diversity of the normal traffic in a honeypot setup is quite restricted. Also, since the labeling is done using existing software and rules, the dataset's anomalies might be underestimated.

**Visualization of the data shifts with PCA**   For completeness, in Fig. 9 we illustrate the distribution shift between years using a PCA visualization of the point clouds associated with each year. We observe similar results as the t-SNE visualization presented in Fig. 4.

**Performance evolution over time**   In Tab. 2 and Fig. 10 we present the full evaluation of considered baseline models on IID, NEAR and FAR splits.

**Training strategies for data shift**   In Tab. 3 we present the full ROC-AUC, and PR-AUC for inliers and outliers, for all three training strategies: iid, finetune and distill.

**BERT for Anomalies**   We propose a simplified BERT architecture for detecting anomalies. The network input is tokenized by a WordLevel tokenizer which obtains tokens for the individual events in a system log sequence and, conversely, for the individual features of network traffic. Therefore, we have fixed-length sequences for Kyoto-2006. We train the BERT model as a Masked Language Model (MLM), using a data collator that randomly masks $p\%$ of the input sequence, by optimizing a cross-entropy loss function between the model predictions at mask positions and the original tokens. We derive a sequence anomaly score by randomly masking $p\%$ of tokens in the sequence and averaging the probabilities of the correct tokens at mask positions given by the classification layer over the vocabulary. The model is not pretrained and consists of two hidden layers of size 120, an intermediate size of 192 and 6 attention heads. It has a hidden dropout and attention dropout probabilities of 0.1, an epsilon of $1e - 12$ for the normalization layer and a 0.02 standard deviation range for the truncated normal weight initialization. Our architecture totals 342135 trainable parameters. For training we mask $p = 15\%$ of the input sequence and at evaluation time, we average over $n = 10$ mask samplings.

$$\hat{w}_j^{\ i} = \begin{cases} w_j, & \text{if } mask_i(\text{j}) = 0 \\ [MASK], & \text{if } mask_i(\text{j}) = 1 \end{cases} \tag{1}$$

We repeat the masking process n times and average over all repeats to improve consistency. The anomaly score formula is depicted in equation 2, where we denote by $P_M$ the classification layer of the model of parameters $\theta_M$, by $Masks_k^p$ the set of random binary masks of length k and mask probability p, where $w_t$ are the initial tokens in the sequence and $\hat{w}_t^{\ i}$ is the j-th token in the sequence under mask i.

$$anomaly\_score([w_1, w_2, ..., w_t]) = \frac{\sum_{i=1..n} \sum_{j=1..t}^{mask_i \sim Masks_t^p} (1 - P(\hat{w}_j^{\ i}))}{n} \tag{2}$$

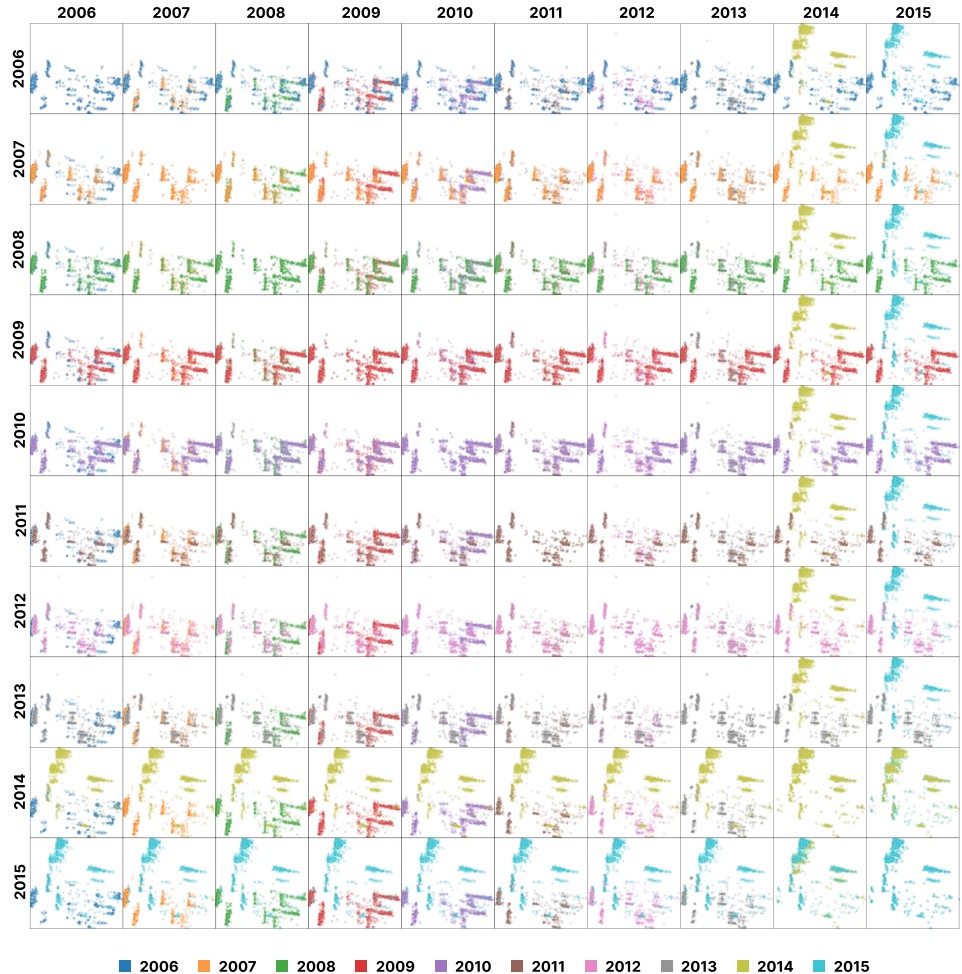

Figure 9: Comparison between yearly splits using PCA visualization. Similar to the t-SNE visualization, we observe that the discrepancy between point clouds increases with the temporal distance between splits, colors becoming more separated over time.

$$P(\hat{w_j}^i) = \begin{cases} 1, & \text{if } mask_i(\text{j}) = 0 \\ P_M(w_j | \theta_M, [\hat{w_1}^i, ..., \hat{w_t}^i]), & \text{if } mask_i(\text{j}) = 1 \end{cases} \tag{3}$$

We motivate this metric with the observation that inlier data should consist of common tokens with a high retrieval probability given by the distribution of the training set, while outliers should usually have either rare tokens or unusual combinations of features, within the context given by the unmasked tokens.

**Impure training**    Till now, we have considered anomaly detection models that were trained solely on the clean data of the `TRAIN` split. Further, we will study the performance evolution between `IID`, `NEAR` and `FAR` when the BERT model is trained on corrupt data containing mislabeled outliers in different percentages. In Fig. 11 we present the ROC-AUC for the 3 testing splits. We observe that the distribution shift is noticeable in the model performance even in this corrupt training setup. This validates the usefulness of the proposed chronological protocol when dealing with potentially mislabeled samples and highlights that the observed degradation over time is not a consequence of such dataset issues.

**Broader Impact**    Our benchmark proposal is tailored for finding intrusions in a computer network (not at the user level, but at the network level), by detecting anomalous traffic, in a more robust way than before, closer to the real scenario. One use-case is in the IT department of an company

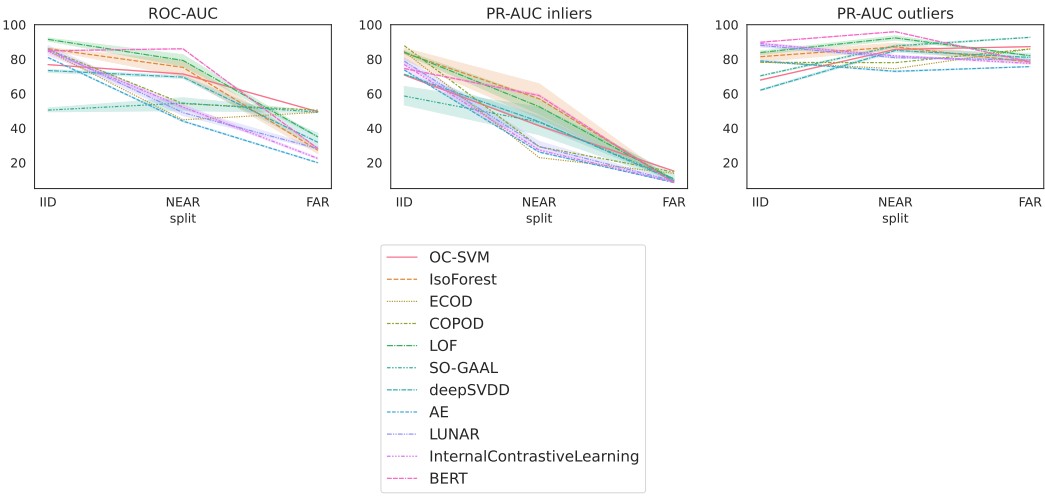

Figure 10: Performance evolution over time: `IID` vs `NEAR` vs `FAR`. We follow the evolution of ROC-AUC and PR-AUC for inliers and outliers. We observe a large performance gap between the considered splits, correlated with the temporal distance from the training set. (Best viewed in color)

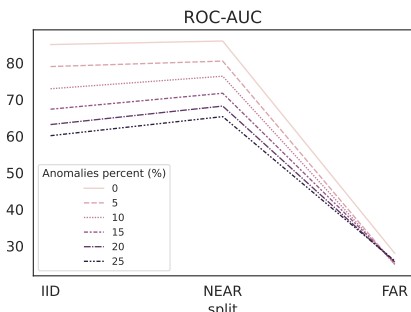

Figure 11: Performance evolution of our BERT model on `IID`, `NEAR` and `FAR` splits, when training on a corrupt set of samples, containing different percentages of mislabeled data points. (Best viewed in color)

or university, where a person monitors the traffic alerts and prioritizes certain alerts based on the predictions of the robust models trained on our proposed benchmark. Our work does not have a negative societal impact.

## A.1    Discussions and future work

**The inliers' natural distribution**    For being able to annotate large amount of data, the network datasets stay either in a clean space, where almost everything is normal, or in a "dark" one, where every connection is considered infected. Kyoto-2006+ lies in the second case, where the normal traffic is not very general, covering several behaviours. It might be interesting as future directions to find a way to combine the two cases towards a more general and unbiased dataset.

**Pre-process through binning**    We have performed the numerical to categorical conversion in order to make the dataset suitable for BERT based models, whose vocabularies would become too large otherwise. For a fair comparison, we consider it proper to use the same preprocessing for all the methods. We binarize 3 numerical features, transforming them into categorical ones (out of 12 total features). Namely, we convert only the float features: connection duration, number of source bytes and number of destination bytes into categorical ones (bins of values). We also perform experiments without preprocessing those numerical features (Tab. 4). Notice that the performance

Table 2: Performance evolution over time for unsupervised methods: `IID` vs `NEAR` vs `FAR`. We report beside the ROC-AUC metric, also the PR-AUC for inliers and PR-AUC for outliers. With bold are the best results per split.

| Type | Unsupervised Baselines | IID | NEAR | FAR |
|---|---|---|---|---|
| | | ROC-AUC (%) ↑ | | |
| Classical | **OC-SVM** [39] (train 5%) | $76.86 \pm 0.06$ | $71.43 \pm 0.29$ | $49.57 \pm 0.09$ |
| | **IsoForest** [27] | $86.09 \pm 0.54$ | $75.26 \pm 4.66$ | $27.16 \pm 1.69$ |
| | **ECOD** [24] | $84.76$ | $44.87$ | $49.19$ |
| | **COPOD** [23] | $85.62$ | $54.24$ | $\mathbf{50.42}$ |
| | **LOF** [5] | $\mathbf{91.50} \pm 0.88$ | $79.29 \pm 3.33$ | $34.96 \pm 0.14$ |
| Deep | **SO-GAAL** [28] | $50.48 \pm 1.13$ | $54.55 \pm 3.92$ | $49.35 \pm 0.51$ |
| | **deepSVDD** [36] | $73.43 \pm 0.94$ | $69.61 \pm 0.83$ | $31.81 \pm 4.54$ |
| | **AE** [1] **for anomalies** | $81.00 \pm 0.22$ | $44.06 \pm 0.57$ | $19.96 \pm 0.21$ |
| | **LUNAR** [14] (train 5%) | $85.75 \pm 1.95$ | $49.03 \pm 2.57$ | $28.19 \pm 0.9$ |
| | **InternalContrastiveLearning** [41] | $84.86 \pm 2.14$ | $52.26 \pm 1.18$ | $22.45 \pm 0.52$ |
| | **BERT [11] for anomalies** | $84.54 \pm 0.07$ | $\mathbf{86.05} \pm 0.25$ | $28.15 \pm 0.06$ |
| | | PR-AUC inliers (%) ↑ | | |
| Classical | **OC-SVM** [39] (train 5%) | $70.84 \pm 0.13$ | $41.38 \pm 0.29$ | $\mathbf{15.12} \pm 0.04$ |
| | **IsoForest** [27] | $83.68 \pm 3.47$ | $57.06 \pm 10.27$ | $9.16 \pm 0.18$ |
| | **ECOD** [24] | $84.47$ | $22.98$ | $13.78$ |
| | **COPOD** [23] | $\mathbf{87.86}$ | $29.25$ | $14.55$ |
| | **LOF** [5] | $84.11 \pm 0.96$ | $52.48 \pm 4.56$ | $10.15 \pm 0.10$ |
| Deep | **SO-GAAL** [28] | $58.65 \pm 5.36$ | $43.52 \pm 11.62$ | $10.68 \pm 2.42$ |
| | **deepSVDD** [36] | $71.24 \pm 0.44$ | $43.80 \pm 2.87$ | $9.72 \pm 0.65$ |
| | **AE** [1] **for anomalies** | $73.76 \pm 0.09$ | $26.16 \pm 0.15$ | $8.51 \pm 0.01$ |
| | **LUNAR** [14] (train 5%) | $78.91 \pm 1.69$ | $29.36 \pm 2.58$ | $9.33 \pm 0.11$ |
| | **InternalContrastiveLearning** [41] | $76.96 \pm 2.12$ | $27.28 \pm 0.59$ | $8.81 \pm 0.05$ |
| | **BERT [11] for anomalies** | $74.61 \pm 0.13$ | $\mathbf{58.94} \pm 0.69$ | $8.22 \pm 0.02$ |
| | | PR-AUC outliers (%) ↑ | | |
| Classical | **OC-SVM** [39] (train 5%) | $67.94 \pm 0.21$ | $85.70 \pm 0.16$ | $87.27 \pm 0.02$ |
| | **IsoForest** [27] | $81.46 \pm 2.52$ | $87.13 \pm 2.08$ | $78.33 \pm 1.41$ |
| | **ECOD** [24] | $78.37$ | $74.48$ | $85.9$ |
| | **COPOD** [23] | $78.19$ | $77.99$ | $85.98$ |
| | **LOF** [5] | $83.86 \pm 0.98$ | $92.34 \pm 1.26$ | $81.99 \pm 0.05$ |
| Deep | **SO-GAAL** [28] | $70.38 \pm 0.28$ | $87.71 \pm 0.74$ | $\mathbf{92.67} \pm 0.13$ |
| | **deepSVDD** [36] | $62.06 \pm 0.42$ | $85.05 \pm 0.86$ | $81.03 \pm 2.31$ |
| | **AE** [1] **for anomalies** | $78.99 \pm 0.28$ | $72.97 \pm 0.38$ | $75.71 \pm 0.05$ |
| | **LUNAR** [14] (train 5%) | $88.01 \pm 1.03$ | $80.91 \pm 0.62$ | $79.45 \pm 0.30$ |
| | **InternalContrastiveLearning** [41] | $89.08 \pm 0.87$ | $81.93 \pm 0.39$ | $77.55 \pm 0.50$ |
| | **BERT [11] for anomalies** | $\mathbf{89.83} \pm 0.07$ | $\mathbf{95.96} \pm 0.06$ | $78.38 \pm 0.02$ |

varies depending on the method and split, and it is not clear that one feature set is better than the other, over all the methods. Nevertheless, we see the same trend of performance drop across the three splits, supporting the claim of our work. Additionally, we observe that the evaluation on data without preprocessing numerical features instead of categorical ones achieves a better score on FAR. This might be explained by the fact that numerical binning induces a higher rarity of tokens in the FAR split compared to IID and NEAR, and therefore resulting in more uncertainty for the models.

**Gap between supervised and unsupervised learning**    We evaluate several supervised learning methods for anomaly detection modeled as a binary classification task, on our AnoShift benchmark. We test several classical baselines: (SVM [9], RandomForest [25], XGBoost [8]), but also some attention-based deep learning methods (BERT with a classification head [10], TabNet [4],

Table 3: Training strategies: ROC-AUC (%) for IID training vs Finetune vs Distil on Kyoto-2006+

| Strategy | Train ON: **2006 ->** Test split | **2007**↑ | **2008**↑ | **2009**↑ | **2010**↑ |
|---|---|---|---|---|---|
| **IID** | 2011 | 88.95 | 88.93 | 88.27 | 89.92 |
| | 2012 | 95.85 | 90.96 | 86.28 | 86.63 |
| | 2013 | 94.05 | 87.00 | 80.79 | 81.87 |
| | 2014 | 28.56 | 24.35 | 22.64 | 21.16 |
| | 2015 | 49.01 | 42.20 | 37.05 | 34.07 |
| **Finetune** | 2011 | 88.56 | 87.18 | 87.3 | 89.92 |
| | 2012 | 95.78 | 89.31 | 85.18 | 89.14 |
| | 2013 | 94.19 | 85.29 | 79.36 | 84.06 |
| | 2014 | 32.98 | 22.50 | 20.50 | 20.57 |
| | 2015 | 53.39 | 38.99 | 30.98 | 30.04 |
| **Distil** | 2011 | 83.74 | 87.43 | 88.82 | 90.32 |
| | 2012 | 95.10 | 91.96 | 88.78 | 91.51 |
| | 2013 | 94.43 | 88.89 | 83.43 | 86.04 |
| | 2014 | 43.65 | 26.69 | 23.55 | 23.13 |
| | 2015 | 59.93 | 48.39 | 41.31 | 39.69 |

Table 4: Compare different numeric feature preprocessing. Notice that it is not clear that one feature set is better than the other, over all the methods. Nevertheless, we see the same trend of performance drop across the splits, supporting the claim of our work.

| Method | Feature binarization | IID | NEAR | FAR |
|---|---|---|---|---|
| | | ROC-AUC (%) ↑ | | |
| OC-SVM [39] | w | 76.86 | 71.43 | 49.57 |
| | w/o | 81.87 | 71.24 | 50.95 |
| IsoForest [27] | w | 86.09 | 75.26 | 27.16 |
| | w/o | 94.41 | 95.13 | 32.81 |
| ECOD [24] | w | 84.76 | 44.87 | 49.19 |
| | w/o | 79.38 | 69.80 | 60.84 |
| COPOD [23] | w | 85.62 | 54.24 | 50.42 |
| | w/o | 79.03 | 65.67 | 60.12 |
| AE [1] | w | 81.00 | 44.06 | 19.96 |
| | w/o | 89.59 | 86.76 | 30.79 |

SAINT [42]). We report in Tab. 5 ROC-AUC, AUC-PR for Inliers and for Outliers for the supervised baselines, where we also included our unsupervised BERT baseline for comparison. We plotted the results in Fig. 12. We observe highly saturated scores on **IID** and **NEAR** and a major performance degradation on **FAR**. The highest performing methods on **IID** and **NEAR**, XGBoost, BERT and Saint, achieve the lowest scores on **FAR** across all baselines.

**Full Kyoto-2006+ dataset**   As previously described, AnoShift contains subsets of the full data, for allowing faster prototyping. We evaluate BERT for anomalies on the full Kyoto-2006+ yearly sets and observe that the ROC-AUC results are consistent with the subsets. The evaluation is performed on held-out test sets for each year and the results are available in Tab. 6. The subsets as well as the full sets used in our experiments are available at `https://share.bitdefender.com/s/9D4bBE7H8XTdYDB`.

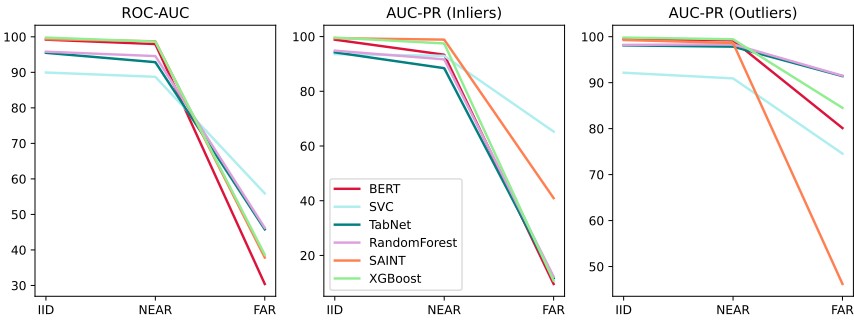

Figure 12: Performance evaluation for several supervised learning baselines in a binary classification task on the Kyoto-2006+ dataset.

## A.2 Pseudo-code for BERT training

We present the three algorithms for each training strategies: IID 1, Finetune 2, and Distilation 3.

---

**Algorithm 1** IID training
---

$Model \leftarrow init\_model()$
$optimizer \leftarrow AdamW()$
$set \leftarrow shuffle(concat(set_1, ..., set_n))$
**for** $epoch \leftarrow [1, ..., num\_epochs]$ **do**
    **for** $batch \sim set$ **do**
        $mask \sim random(batch.shape) < 0.15$   ▷ Sample a binary mask of batch size with 0.15
probability
        $predictions = Model(batch * mask)$
        $loss\_batch = loss(predictions, batch)$        ▷ Reconstruction loss for masked tokens
        compute loss gradients
        perform optimizer step
    **end for**
**end for**

---

---

**Algorithm 2** Finetune strategy
---

$Model \leftarrow init\_model()$
$loss \leftarrow CrossEntropy()$
$optimizer \leftarrow AdamW()$
**for** $set \leftarrow [set_1, ..., set_n]$ **do**
    **for** $epoch \leftarrow [1, ..., num\_epochs]$ **do**
        **for** $batch \sim set$ **do**
            $mask \sim random(batch.shape) < 0.15$   ▷ Sample a binary mask of batch size with
0.15 probability
            $predictions = Model(batch * mask)$
            $loss\_batch = loss(predictions, batch)$     ▷ Reconstruction loss for masked tokens
            compute loss gradients
            perform optimizer step
        **end for**
    **end for**
**end for**

---

## A.3 Other considered datasets

We show here the detailed process of how we choose the Kyoto-2006+ dataset and why we consider it to be one of the few relevant in the distribution-shift context for stream-like data. We performed an

Table 5: Performance evolution over time for supervised methods: `IID` vs `NEAR` vs `FAR`. We report the ROC-AUC, PR-AUC for inliers, and PR-AUC for outliers metrics. The performance degrades over time also in this supervised setting. Notice there is a large (and consistent) gap between the supervised methods and the unsupervised BERT baseline. Best score per split in bold.

| Supervised Baselines | IID | NEAR | FAR |
|---|---|---|---|
| | ROC-AUC (%) ↑ | | |
| XGB [8]) | **99.79** ± 0.01 | 98.63 ± 0.03 | 38.66 ± 0.32 |
| SVM [9] | 89.95 ± 0.81 | 88.74 ± 0.51 | **55.92** ± 0.21 |
| RandomForest [25] | 95.81 ± 0.13 | 94.58 ± 0.40 | 46.22 ± 0.73 |
| SAINT [42]) | 99.33 ± 0.10 | **98.74** ± 0.09 | 37.85 ± 0.65 |
| TabNet [4] | 95.49 ± 0.28 | 92.86 ± 0.64 | 45.80 ± 0.57 |
| BERT [10] - sup | 99.20 ± 0.02 | 97.96 ± 0.08 | 30.42 ± 0.40 |
| BERT [10] - unsup | 84.54 ± 0.07 | 86.05 ± 0.25 | 28.15 ± 0.06 |
| **Difference** (best sup, BERT-unsup) | +15.25 | +12.69 | +27.77 |
| | PR-AUC Inliers (%) ↑ | | |
| XGB [8]) | **99.64** ± 0.01 | 97.49 ± 0.02 | 10.72 ± 0.05 |
| SVM [9] | 93.57 ± 0.10 | 92.96 ± 0.20 | **65.27** ± 0.42 |
| RandomForest [25] | 94.85 ± 0.13 | 91.67 ± 0.91 | 12.27 ± 1.52 |
| SAINT [42]) | 99.41 ± 0.08 | **98.89** ± 0.10 | 40.95 ± 0.42 |
| TabNet [4] | 94.24 ± 0.87 | 88.45 ± 1.89 | 11.63 ± 0.72 |
| BERT [10] - sup | 98.87 ± 0.04 | 93.37 ± 0.19 | 9.57 ± 0.06 |
| BERT [10] - unsup | 74.61 ± 0.13 | 58.94 ± 0.69 | 8.22 ± 0.02 |
| **Difference** (best sup, BERT-unsup) | +25.03 | +39.95 | +57.05 |
| | PR-AUC Outliers (%) ↑ | | |
| XGB [8]) | **99.81** ± 0.01 | **99.44** ± 0.02 | 84.52 ± 0.21 |
| SVM [9] | 92.16 ± 0.22 | 90.93 ± 0.39 | 74.56 ± 0.14 |
| RandomForest [25] | 98.21 ± 0.06 | 98.34 ± 0.11 | **91.51** ± 0.12 |
| SAINT [42]) | 99.28 ± 0.11 | 98.63 ± 0.07 | 46.21 ± 1.11 |
| TabNet [4] | 98.11 ± 0.04 | 97.83 ± 0.16 | 91.42 ± 0.27 |
| BERT [10] - sup | 99.51 ± 0.02 | 99.20 ± 0.04 | 80.12 ± 0.07 |
| BERT [10] - unsup | 89.83 ± 0.07 | 95.96 ± 0.06 | 78.38 ± 0.02 |
| **Difference** (best sup, BERT-unsup) | +9.98 | +3.48 | +13.13 |

Table 6: BERT for anomalies ROC-AUC evaluation on the full sets in comparison with the subsets

| Split | ↑ ROC-AUC | | | | | | | | | |
|---|---|---|---|---|---|---|---|---|---|---|
| | **2006** | **2007** | **2008** | **2009** | **2010** | **2011** | **2012** | **2013** | **2014** | **2015** |
| Full | 83.07 | 84.84 | 82.39 | 85.87 | 84.98 | 90.79 | 90.40 | 86.74 | 24.05 | 38.84 |
| 300k Subset | 82.20 | 84.63 | 83.80 | 85.60 | 83.51 | 88.03 | 88.12 | 82.31 | 22.10 | 36.90 |

in depth analysis over a large number of datasets. We wanted it to come from a stream-like data (as opposed to the less natural, existing benchmarks on images or text [26, 22, 44, 6, 20] and we for two **characteristics that we consider essential for a distribution shift benchmark**:

- It spreads over a large enough time-span, such that the distribution shift will naturally occur, (rather than being synthetically injected, exhibiting sudden changes)

- It is not solved already (existing methods do not report almost perfect scores on it)

**Network traffic datasets** We first looked over a wide range of known network traffic datasets for intrusion detection (see Tab. 7), and after analysing them we concluded that most are artificially

**Algorithm 3** Distillation strategy

---

$mlm\_loss \leftarrow CrossEntropy()$
$distil\_loss \leftarrow KL\_divergence()$
$optimizer \leftarrow AdamW()$
$Teacher \leftarrow$ train IID on $set_1$
**for** $set \leftarrow [set_2, ..., set_n]$ **do**
    $Student \leftarrow init\_model()$
    **for** $epoch \leftarrow [1, ..., num\_epochs]$ **do**
        **for** $batch \sim set$ **do**
            $mask \sim random(batch.shape) < 0.15$   ▷ Sample a binary mask of batch size with 0.15 probability
            $pred\_s = Student(batch * mask)$
            $pred\_t = Teacher(batch * mask)$
            $loss\_batch = mlm\_loss(pred\_s, batch) + distil\_loss(pred\_s, pred\_t)$
            compute loss gradients
            perform optimizer step
        **end for**
    **end for**
    $Student \leftarrow Teacher$
**end for**

---

Table 7: Network traffic datasets.

| Dataset | Number of samples | Time-span | Other details |
|---|---|---|---|
| CIC-IDS2017 | 3 mil | 5 days | Different attack types per day |
| CSE-CIC-IDS2018 | 4.5 mil | 17 days | Different attack types per day |
| UNSW-NB15 | 2.5 mil | 2 days | too small |
| BoT-IoT | 73 mil | 4 days | too small |
| ToN-IoT | 22 mil | 6 days | too small |
| NSL-KDD | 0.15 mil | 45 days | max reported ROC-AUC 99% |
| LANL | 1.6 mil | 58 days | max reported ROC-AUC 99% |
| AAD | 1.8 mil | 90 days | internally build dataset, max ROC-AUC 98% |
| Kyoto-2006+ | 806M | 10 years | |

created, with injected samples, in very restricted scenarios. Only Kyoto-2016 was a proper dataset, extended over a long enough period of time for showing a natural distribution shift.

**System logs datasets**   We next focused our attention on system logs, since the time-span is usually more extensive in these dataset and the natural distribution shift is more probable to occur. But under our analysis (t-SNE, Jeffreys divergence, OTDD, multiple baselines), these datasets did not exhibit a clear distribution shift over time, so we decided to further analyse them until concludent results. We used Drain and Spell as log parsers, and we report in Tab. 8 the results using the LogAnomaly [30] baseline.

**Multi-variate timeseries datasets**   We next looked over general multi-variate timeseries datasets, but the most popular ones are quite small and almost perfectly solved already (see Tab. 9).

Table 8: System logs datasets.

| Dataset - Preprocessor | Number of samples | Time-span | IID (%) | NEAR (%) | FAR (%) | Split proportion |
|---|---|---|---|---|---|---|
| HDFS - Drain | 11 mil | 40h | 54 | 66 | 57 | 6-6-6 |
| BGL - Spell/Drain | 4.7 mil | 214 days | 67/68 | 43/73 | 45/35 | 2-3-2 |
| Thunderbird - Spell/Drain | 211 mil | 244 days | 72/71 | 72/72 | 76/75 | 3-3-3 |
| Liberty | 266 mil | 315 days | | | | grouped anomalies |
| Spirit-CMU - Spell | 272 mil | 570 days | 80 | 67 | 72 | 6-5-3 |

Table 9: Multi-variate timeseries datasets.

| Dataset | Number of samples | Time-span | max reported unsup ROC-AUC (%) |
|---|---|---|---|
| SMAP - Soil Moisture Active Passive | 0.5 mil | 7-14 days | 99 |
| SWaT - Secure Water Treatment | 0.9 mil | 11 days | 85 |
| WADI - Water Distribution | 0.96 mil | 16 days | 90 |
| SMD - Server Machine Dataset | 1.4 mil | 35 days | 99 |
| MSDS - Multi-Source Distributed System | 0.3 mil | days - months | 91 |
| PSM - Pooled Server Metrics | | 147 days | 98 |
| MSL - Mars Science Laboratory | 0.13 mil | - | 99 |
| NAB - Numenta Anomaly Benchmark | 0.37 mil | - | 99 |
| MBA - MIT-BIH Supraventricular Arrhythmia | 0.2 mil | 78 half-hour ECGs | 99 |

# B    Appendix

**Raw Kyoto dataset documentation**    The dataset used in our proposed benchmark consists of a preprocessing of the Traffic Data from Kyoto University's Honeypots and results from a discretization of the numerical features in the original dataset, such that a language-modelling approach can be easily applied. The original dataset consists of 14 conventional features and 10 additional features. The conventional features in the original dataset includes connection duration, type of service, number of source and destination bytes, server rate errors percentage and flag of connection. We keep all the conventional features and apply a exponentially-scaled binning over the continuous values (duration, number of source and destination bytes) which results in 233 bins and a discretization of the percentage features in 100 distinct values. As an observation, some of the 10 additional features (the source and destination IP addresses, source and destination port numbers) might be useful when designed models (eg. graphs) focusing on connections between the nodes in the system.

**Our split proposal documentation**    We propose a yearly split of the dataset and group adjacent years into Train, NEAR data and FAR data, which we use to highlight the performance degradation of several benchmarks in time, due to the distributional shift of the data which we demonstrate with a comprehensive analysis. In our proposed split, Train data consists of the first 4 years (2006-2010), Near data of the following 3 years (2011-2013) and Far data of the last two available years (2014-2015). We publish the data in splits of single years, in csv format. The columns 0 to 13 are the discretized conventional features in the Kyoto-2006+ dataset, preserving the original order, column 14 contains the complete timestamp. Columns 15, 16 and 17 correspond to the first 3 additional features in the original data, namely IDS_detection, Malware_detection and Ashula_detection, which indicates presence of alert triggers from the 3 IDS solutions: Symantec IDS, clamav and Ashula shellcode detector. Column 19 in the preprocessed dataset corresponds to the protocol used by the connection.

**Intended uses**    We hope that our proposed benchmark shifts the general direction of treating network intrusion detection towards a timely fashion that suffers from distributional shift, hereby providing a better suited evaluation protocol for upcoming research in this field.

**URL to dataset download**    We redirect our readers to the repository of the raw Kyoto dataset published by the Kyoto University at `https://www.takakura.com/Kyoto_data/` and provide a repository of data under our proposed processing at `https://share.bitdefender.com/s/9D4bBE7H8XTdYDB`, with subsets of 300000 instances and heldout sets of 30000 instances for each split, maintaining the original inlier to outlier ratio from the original data in each split, as well as the full processed splits, with each full split except 2006 being provided in two parts. We make the remark that the 2006 split contains fewer instance, due to data collection debuting in November.

## B.1    Code for dataset loading

We publish our code as a public GitHub repository `https://github.com/bit-ml/AnoShift/`, containing the data preprocessing script that transforms the original data in our format, sample data manipulation notebooks, license and additional information.

## B.2    Author responsibility for violation of rights

There is no sensitive data leaked in the preprocessed dataset. The authors are not aware of any possible violation of rights and take responsibility for the published data.

## B.3    Dataset hosting and long-term preservation

The authors take full responsibility for the availability of the processed data in the provided repository. However, no statement can be made about the availability of the raw Kyoto-2006+ data published by the Kyoto University, as it depends on Takakura.com. To avoid further problems, we have published our preprocessed version.

## B.4 Licence

We release our code under a BSD 3-Clause License, therefore allowing the redistribution and use in source and binary forms, with or without modification, under the 3 clauses specified by the Berkeley Software Distribution License:

1. Redistributions of source code must retain the above copyright notice, this list of conditions and the following disclaimer.

2. Redistributions in binary form must reproduce the above copyright notice, this list of conditions and the following disclaimer in the documentation and/or other materials provided with the distribution.

3. Neither the name of the copyright holder nor the names of its contributors may be used to endorse or promote products derived from this software without specific prior written permission.