# OpenReview forum: "AnoShift: A Distribution Shift Benchmark for Unsupervised Anomaly Detection"
_NeurIPS.cc/2022/Track/Datasets_and_Benchmarks — NeurIPS 2022 Datasets and Benchmarks _

### Official Review · Reviewer_VE9Z · 2022-07-07
**In this paper, the authors analyze the distribution shift in the Kyoto-2006+ network traffic dataset from different perspectives using the t-SNE and Optimal Transport Dataset Distance method (OTDD). They further propose an unsupervised anomaly detection benchmark, AnoShift, and validate the performance improvement by acknowledging the distribution shift problem and properly addressing it.**

**Rating:** 7
**Confidence:** 4
**Correctness:** Yes.

**Strengths:**

1) The exploration of this benchmark in unsupervised learning is encouraging since such works can reduce the time and money burden brought by labeling data.

2)  The idea of splitting the test dataset into three splits according to their temporal distance to the training set is straightforward and reasonable. This benchmark provides a close-to-real-world way to evaluate the performance and generalization of a trained model.

3) The distribution shift problem is analyzed via different visual experiments and the performance improvement of a model by solving this problem is verified via corresponding experiments, which gives some valuable insights into the dataset and benchmark community.

**Weaknesses:**

1) The writing of this paper is not standardized enough. For example, the terms or abbreviations appearing for the first time in a paper should be explained, but I don’t know the meaning of ‘IID’ and other words. In addition, there are some ill sentences in this paper, such as the 56-th row.

2) This paper lacks the comparative experiments between the unsupervised benchmark proposed in the paper and the existing supervised methods. It is really valuable to clarify the gap between unsupervised and supervised methods and encourage more people to reduce this gap. I think the authors can consider adding this experiment as the future work of this paper.

3) The explanation of some experimental results in this paper is not clear. Take Tab.1 as an example. Why the performance of the transformer model proposed in this paper (BERT for anomalies) in the IDD split is lower than the one in the NEAR split? Why the performance of this model is lower than other compared models except IsoForest in FAR split?

**Additional Feedback:**

No.

**Clarity:**

The paper is overall well-motivated and easy to follow. And I think it would be better if the authors corrected this mistake in the first weakness I propose.

**Documentation:**

Yes, the authors provide the dataset and code via a Github link: https://github.com/bit-ml/AnoShift/.


**Ethics:**

No.

**Relation To Prior Work:**

In Section 2, the authors discuss the relation and difference between this benchmark to other benchmarks targeting distribution shift. They also discuss the relation anomalies.

**Summary And Contributions:**

1)  this paper analyzes Kyoto-2006+, which is a network traffic dataset for unsupervised anomaly detection tasks, and demonstrates that it is affected by distribution shifts.

2) this paper proposes a chronology-based benchmark to enable a better estimate of the model’s performance in an environment close to the real world.  This benchmark splits the test data based on its temporal distance to the training set and introduces three testing splits IID, NEAR, and FAR.

3) this paper proves that by acknowledging the distribution shift problem and properly addressing it, the performance can be improved compared to the classical IID training.

---

> ### Author Response · Authors · 2022-08-16
> **Response to  Reviewer VE9Z - Part1**
>
> Thank you for your observations and proposals to improve our paper. We individually address them below:
>
> ---
>
> **Q1.** **[Clarity]** The writing of this paper is not standardized enough. For example, the terms or abbreviations appearing for the first time in a paper should be explained, but I don’t know the meaning of ‘IID’ and other words. In addition, there are some ill sentences in this paper, such as the 56-th row.
>
> **R1.** We are deeply sorry for the misunderstanding. We have **properly introduced the abbreviations in our revised paper**. We considered three splits for our dataset: IID, NEAR, FAR. The **IID abbreviation** indicates the standard assumption we make when training deep learning models, when we **consider that both testing and training data come from the same distribution**. Our proposed benchmark emphasizes the natural in-time distribution shift of network traffic. We introduce the NEAR and FAR testing splits, where NEAR indicates the split closer (from a temporal point of view) to the training set, while FAR is the one farther away. The whole point is to have a better understanding of the real-world performance of anomaly detection algorithms, as they are expected to face distribution shifts (as the ones between IID, NEAR, and FAR splits) when deployed in real case scenarios.
>
> ---
>
> **Q2**. **[Supervised vs Unsupervised]** This paper lacks the comparative experiments between the unsupervised benchmark proposed in the paper and the existing supervised methods. It is really valuable to clarify the gap between unsupervised and supervised methods and encourage more people to reduce this gap. I think the authors can consider adding this experiment as the future work of this paper.
>
> **R2**. Thank you, this is an excellent suggestion and a relevant analysis that was missing from our study. We performed supervised learning experiments with several baselines (BERT for text classification, SVM, Random Forest, XGBoost, TabNet, SAINT) in a binary classification task and observed that the best performing models saturate quite fast on the IID split, while obtaining high-performance on NEAR. Even in the supervised setup, the performance on the FAR split drops significantly and the distribution shift we observe is consistent with the observations from the unsupervised scenario proposed in AnoShift. We added the experimental results in Appendix A.1, Tab. 5. In the following table, we show ROC-AUC values. To highlight the **gap between supervised and unsupervised methods**, we have also included the results of our unsupervised BERT baseline.
>
> |          	|  IID | NEAR |  FAR |
> |--------------|:----:|:----:|:----:|
> | XGB      	| 99.8 | 98.6 | 38.8 |
> | SVM      	| 90.4 | 89.1 | 55.8 |
> | TabNet   	| 95.7 | 93.1 |   45 |
> | RandomForest | 95.8 | 95.3 | 47.5 |
> | SAINT    	| 99.4 | 98.7 | 38.4 |
> | BERT-sup 	| 99.2 |   98 | 30.1 |
> | BERT-unsup   | 84.5 | 86.1 | 28.2 |
>
>
> [BERT] Devlin J, Chang MW, Lee K, Toutanova K. Bert: Pre-training of deep bidirectional transformers for language understanding, NAACL 2019
>
> [SVM] Cortes C, Vapnik V. Support-vector networks, Machine learning 1995
>
> [RandomForest] Liaw A, Wiener M. Classification and regression by randomForest, R news 2002
>
> [XGB] Friedman JH. Greedy function approximation: a gradient boosting machine, Annals
> of Statistics 2001
>
> [TabNet] Arik SÖ, Pfister T. Tabnet: Attentive interpretable tabular learning, AAAI 2021
>
> [SAINT] Somepalli G, Goldblum M, Schwarzschild A, Bruss CB, Goldstein T. Saint: Improved neural networks for tabular data via row attention and contrastive pre-training, arXiv 2021
>
> ---
>
> (continues in Part2)

---

> > ### Author Response · Authors · 2022-08-16
> > **Response to  Reviewer VE9Z - Part2**
> >
> > **Q3.** **[BERT - NEAR vs IID]** The explanation of some experimental results in this paper is not clear. Take Tab.1 as an example. Why the performance of the transformer model proposed in this paper (BERT for anomalies) in the IID split is lower than the one in the NEAR split? Why the performance of this model is lower than other compared models except IsoForest in FAR split?
> >
> > **R3.** We thank the reviewer for pointing this out. We further present 3 different points of view for addressing it. We added this discussion in the main paper and clarify in Tab. 1 caption:
> >
> > a. For better understanding this result, we further tested **more baselines** (both classical and new methods - Tab. 1). We observe that **in general, the scores are decreasing from IID to NEAR**, for almost all tested methods (9 out of 11, best viewed in Fig. 11 - Appendix). But BERT and the SO-GAAL are exceptions. We argue that this happened due to certain method particularities, rather than the dataset (also motivated by our t-SNE, Jeffreys divergence, and OTDD analysis). Moreover, the ROC-AUC evolution for the two methods is explained by a very large PR-AUC-outliers compared with others, while the PR-AUC-inliers follows the gradual degradation over time (Tab. 2 in Appendix). To emphasize the differences, we also add the best score per split (different from BERT or SO-GAAL).
> >
> > | ROC-AUC | Method  | IID| NEAR| FAR|
> > | ------------- |:-------------| ------------- |:-------------:|:-------------:|
> > || **BERT** | 84.54%| 86.05% | 28.15%|
> > || **SO-GAAL**| 61.82%| 81.24% |39.74%|
> > || **(different) Best per split** | 86.58 %| 74.45% | 46.96 %|
> >
> >
> >
> > | PR-AUC-in | Method  | IID| NEAR| FAR|
> > | ------------- |:-------------| ------------- |:-------------:|:-------------:|
> > || **BERT** |74.61%|58.94% |8.22%|
> > || **SO-GAAL** |61.53%|52.66% |9.97%|
> > || **(different) Best per split** |78.77%|44.71% |13.14%|
> >
> >
> >
> > | PR-AUC-out | Method  | IID| NEAR| FAR|
> > | ------------- |:-------------| ------------- |:-------------:|:-------------:|
> > || **BERT** |89.83%|95.96% |78.38%|
> > || **SO-GAAL** |65.53%|93.41% |88.59%|
> > || **(different) Best per split** |88.81%| 89.60% |86.80%|
> >
> >
> >
> > b. We have an additional confirmation for the distribution shift in AnoShift setting, by testing several **supervised methods**, and we see there that **the performance also degrades over time** (see Tab. 5 and Fig. 12 in the Appendix).
> >
> > c. We also present **some insights** which we consider that might **prevent BERT** (or Masked Language Models in general) **to be a robust anomaly detector**, and we think that might be interesting for future research:
> > * **MLM as anomaly detector**: Even with a large number of parameters, the anomaly performance of BERT is based on the perplexity score when predicting several masked features in the sample. So if the features are not correlated, the MLM model might be unable to learn something useful, which might result in learning some specific training set biases, failing to generalize on further data (eg. lower score on FAR wrt other baselines).
> > * **MLM with the training vocabulary**: If the model only learns based on the training data, all the new words (in our case, features from a new sample) will be considered UNK, and the score might get artificially inflated (since from a point further it is easier to predict UNK instead of the right word). This might be a disadvantage of vocabulary-based methods as opposed to other classic approaches.
> >
> > [SO-GAAL] Yezheng Liu, Zhe Li, Chong Zhou, Yuanchun Jiang, Jianshan Sun, Meng Wang, and Xiangnan He. Generative adversarial active learning for unsupervised outlier detection. IEEE Transactions on Knowledge and Data Engineering, 2019.
> >
> > ---

---

> > > ### Comment · Reviewer_VE9Z · 2022-08-24
> > > **Good response**
> > >
> > > I would like to increase my rating after reading the authors' responses.

---

### Official Review · Reviewer_KLv6 · 2022-07-15
**Interesting Dataset with Naturally Occuring Distribution Shift**

**Rating:** 7
**Confidence:** 3

**Strengths:**

* Understanding distribution shift (particularly in the anomaly detection literature) is relatively under-explored. The proposed Kyoto-2006+ dataset is an important step in this direction.

* The Kyoto-2006+ dataset covers a span of 10 years with naturally occuring distribution shift. The authors also empirically validated the performance degradation of models due to distribution shift using proposed split strategies.

* Dataset statistics and visualizations are explored extensively in Section 4 to analyze the distribution shift.

* data is open sourced already and can be easily downloaded / loaded into ML pipeline

**Weaknesses:**

* In the preprocessing stage, numerical features are converted to categorical ones to use in language modelling. Is this performed for all methods? How would the performance be without this conversion (using the numerical values directly)?

* Is there a reason in Section 4.1 that only t-SNE visualization is chosen? How would Figure 4 look like with a PCA plot (with just 2d PCA).

* Network intrusion detection has often been studied in literature as a graph anomaly detection problem, for example in [Fast and Accurate Anomaly Detection in Dynamic Graphs
with a Two-Pronged Approach](https://arxiv.org/pdf/2011.13085.pdf) and [SEDANSPOT: Detecting Anomalies in Edge Stream](https://dhivyaeswaran.github.io/papers/icdm18-sedanspot.pdf). The DARPA dataset is a well-known network intrusion detection dataset formulated as a dynamic graph (see above two references). Have the authors considered formulating this dataset as a dynamic graph too? Are there limitations in the data which restricts it from being a graph dataset? If not, it is encouraged that the authors also considers converting the dataset into a graph dataset for use in the graph anomaly detection community.

*In the appendix it was mentioned that "we discard the source and destination IP addresses, source and destination port numbers." Is there a specific reason for doing this? With proper anonymization, these features can also be kept and used in a graph format.

**Additional Feedback:**

I ask the authors to address limitations from the weakness section and in particular, examining if a dynamic graph format of the dataset can be provided as it would be valuable to the temporal graph / dynamic graph learning community.

[update] The authors have addressed my concerns and I believe this paper will be a valuable contribution in studying distribution shift

**Clarity:**

Yes, the paper is relatively well-written. Some clarification such as those in weakness section can help improve the writing of the paper.

**Correctness:**

The claims made in the submission are correct. The dataset is constructed in a sound way and experiments performed correctly.

**Documentation:**

Yes, all of these have been addressed and data preprocessing can also be reproduced.

**Ethics:**

No ethical concerns are raised.

**Relation To Prior Work:**

Yes, most of the related work are discussed clearly.
More discussion on dynamic graph anomaly detection methods would be helpful. See [Fast and Accurate Anomaly Detection in Dynamic Graphs
with a Two-Pronged Approach](https://arxiv.org/pdf/2011.13085.pdf) and [SEDANSPOT: Detecting Anomalies in Edge Stream](https://dhivyaeswaran.github.io/papers/icdm18-sedanspot.pdf).

**Summary And Contributions:**

In this work, the authors proposed the first unsupervised anomaly detection benchmark with naturally occuring distribution shift. The authors highlighted the non-stationary nature of the data using per feature analysis, t-SNE plots and an Optimal Transport approach. A novel protocal splitting the data called AnoShift is also proposed which splits the data in IID, NEAR and FAR testing splits. Empirically, it is observed that more severe performance degradation occurs in the FAR split than NEAR split than IID split. Future methods can evaluate upon all three splits in increasing difficulty. The dataset and code are available and pro-processed version of the data is also easily accessible.

---

> ### Author Response · Authors · 2022-08-16
> **Response to Reviewer KLv6 - Part1**
>
> Thank you very much for your valuable comments. Below we address your points individually.
>
> ---
>
> **Q1**.  **[Preprocessing]** In the preprocessing stage, numerical features are converted to categorical ones to use in language modelling. Is this performed for all methods? How would the performance be without this conversion (using the numerical values directly)?
>
> **R1**. This is a fair question, we thank the reviewer for pointing it out. We have performed the numerical to categorical conversion in order to make the dataset suitable for BERT-based models, whose vocabularies would become too large otherwise. For a fair comparison, we consider it proper to **use the same preprocessing for all the methods**. As we mention in Sec. 3.2.1, we binarize 3 numerical features, transforming them into categorical ones (out of 12 numerical features). Namely, we convert only the float features: connection duration, number of source bytes and number of destination bytes into categorical ones (bins of values).
>
> To address the question, we also perform experiments **without preprocessing those numerical features**. We see that the performance varies depending on the method and split, and it is not clear that one feature set is better than the other, over all the methods. Nevertheless, we see the same trend of performance drop across the three splits, supporting the claim of our work. Additionally, we observe that the evaluation on data without preprocessing numerical features instead of categorical ones achieves a better score on FAR. This might be explained by the fact that numerical binning induces a higher rarity of tokens in the FAR split compared to IID and NEAR, and therefore resulting in more uncertainty for the models. Thank you for pointing out this experiment, we added it in the Appendix - Tab. 4.
>
>
> | Method  |IID| NEAR| FAR|
> | ------------- |:-------------| ------------- |:-------------:|
> | **deepSVDD**: as in paper |**69.01**|	**60.35**|	30.89|
> | --------- w/o numerical preprocessing | 65.33	|63.53	|**50.10**|
> ||
> | **LOF**: as in paper |**86.58**	|**74.45**|	32.74|
> | --------- w/o numerical preprocessing | 72.28	| 72.17 | **49.90**|
> ||
> | **OC-SVM**: as in paper |76.78	|**72.73**|	46.96|
> | ---------- w/o numerical preprocessing | **80.69**	|70.73|	**46.24**|
> ||
> | **IsoForest**: as in paper | 78.73 | 58.08 | 26.54  |
> | ---------- w/o numerical preprocessing | **82.66** | **81.68** | **61.96**|
> ||
> | **BERT**: as in paper |84.54	| 86.05 | 28.15 |
> | --------- w/o numerical preprocessing | could not be evaluated since we get a too large vocabulary size|
>
>
> ---
>
> **Q2**. **[PCA visualization]** Is there a reason in Section 4.1 that only t-SNE visualization is chosen? How would Figure 4 look like with a PCA plot (with just 2d PCA).
>
> **R2**. We have chosen t-SNE instead of PCA considering it to be more suitable for our problem as it is a non-linear dimensionality reduction technique. Yet, we can **highlight the same distribution shift using 2D PCA projections**. For completeness, we have added the **PCA visualization** in the **Appendix of our paper**.
>
> ---
>
> **Q3**. **[AnoShift for graph approaches]** Network intrusion detection has often been studied in literature as a graph anomaly detection problem, for example in Fast and Accurate Anomaly Detection in Dynamic Graphs with a Two-Pronged Approach and SEDANSPOT: Detecting Anomalies in Edge Stream. The DARPA dataset is a well-known network intrusion detection dataset formulated as a dynamic graph (see above two references). Have the authors considered formulating this dataset as a dynamic graph too? Are there limitations in the data which restricts it from being a graph dataset? If not, it is encouraged that the authors also considers converting the dataset into a graph dataset for use in the graph anomaly detection community.
>
> **R3**. We highly appreciate your suggestion to expand the usefulness of our benchmark. It is feasible to add this graph structure information to the benchmark by including some additional features (the source and destination IPs and ports, which could be used for building a dynamic graph capturing the traffic evolution). We are confident that the graph structure should reflect the distribution shift between different periods of time, building a very challenging task. It is a great idea, we will consider it as a future extension for AnoShift.
>
> ---
> (continues in Part2)

---

> > ### Author Response · Authors · 2022-08-16
> > **Response to Reviewer KLv6 - Part2**
> >
> > **Q4**. **[Discarded features]** In the appendix it was mentioned that "we discard the source and destination IP addresses, source and destination port numbers." Is there a specific reason for doing this? With proper anonymization, these features can also be kept and used in a graph format.
> >
> > **R4**. Indeed, those features are useful when designing models focusing on connections between system nodes, following connection-level or structure-level anomalies (e.g., graphs). With the current AnoShift benchmark, we did not focus on this class of representations, so we focused our in-depth analysis on the standard features employed in Kyoto-2006+, which did not include IPs, ports, timestamps, and protocols (those are so-called additional features). Including them would have implied a detailed supplementary analysis of those links between the nodes, which would have been too much to adequately cover in a single paper. We clarified this in the paper (Appendix B - first paragraph). For a future update and to make the benchmark more useful and popular, we will consider your idea for expanding the benchmark towards a more general class of methods that allows modeling inter-nodes connections.

---

> > > ### Comment · Reviewer_KLv6 · 2022-08-23
> > > **Response to Authors of Paper 424**
> > >
> > > I thank the authors for addressing my questions and adding additional experiments. The added results are quite interesting.
> > > I believe this work will be a valuable contribution in studying distribution shift and I urge the authors to add the dynamic graph representation of the dataset to the benchmark in the future as it would be valuable for the dynamic graph learning community as well.
> > > I have raised my score to reflect the changes.

---

### Official Review · Reviewer_7yuQ · 2022-07-20
**A good dataset focusing on data distribution shift**

**Rating:** 7
**Confidence:** 4
**Clarity:** Yes.

**Strengths:**

1. The presentation of the paper is clear and easy to follow.

2. The extensive analysis of the dataset is informative, which helps the readers and future benchmark users understand the dataset.

3. The experiments of anomaly detection baselines and the mitigation strategies are strong evidence to support the significance of the proposed benchmark.

**Weaknesses:**

The anomaly is usually defined as data samples that differ from most data samples. However, the proposed dataset has 89.5% anomalies, and I think the task is more like a network intrusion detection task instead of a typical anomaly detection problem. Since the authors mention that Kyoto-2006+ is a classic dataset, the authors should give a comprehensive review of related work using Kyoto-2006+, and it would help me to understand how previous works use the dataset and the contribution significance of the proposed dataset in this paper. Hope the authors can provide more information regarding my concerns above.



**Additional Feedback:**

1. The authors propose a BERT-based anomaly detector, a novel approach to detecting tabular data anomalies. It would be better if the author could open-source the code to facilitate future research.

2. The authors claim the paper is a benchmark paper, but the GitHub repo only has the code for the IF baseline, and the paper does not have the implementation details and parameters of the baselines required by the NeurIPS benchmark track CFP. I suggest the authors either add the missing information to make the paper a benchmark one or rewrite the paper to a dataset paper.

3. Figure 4 shows the distribution shift intuitively, but it takes me some time to interpret, I suggest the authors explain the differences between rows and columns.

4. Please respond to my concerns mentioned in the Weakness section.



**Correctness:**

Though the paper title includes the benchmark, I regard this paper as a dataset paper because the GitHub repo does not include all baselines evaluated on the dataset. As a dataset, it is constructed in a sound way with detailed information on how data is preprocessed.

**Documentation:**

The dataset documentation includes sufficient details in the submitted supplementary materials.

**Ethics:**

No.

**Relation To Prior Work:**

The authors have discussed the previous distribution shift dataset and traffic anomaly detection dataset but miss the literature review of works that use the Kyoto-2006+ dataset.

**Summary And Contributions:**

The authors curated an anomaly detection dataset from a popular network intrusion detection dataset named Kyoto-2006+ and the proposed dataset focus on the distribution shift of the data over time. Extensive analysis of the dataset demonstrates its non-stationary property from multiple perspectives. Some preliminary experiments also show the performance degradation of classic anomaly detection algorithms due to the data distribution shift. Last, the authors compare different continual learning strategies for mitigating model performance degradation on the proposed benchmark and claim the necessity to address the distribution shift.

---

> ### Author Response · Authors · 2022-08-16
> **Response to Reviewer 7yuQ - Part1**
>
> Thank you for your suggestions that have helped us improve both the manuscript and our [project’s repository](https://github.com/bit-ml/AnoShift/). Below are our detailed comments:
>
> ---
>
> **Q1.** **[Kyoto-2006+ discussion]** The anomaly is usually defined as data samples that differ from most data samples. However, the proposed dataset has 89.5% anomalies, and I think the task is more like a network intrusion detection task instead of a typical anomaly detection problem. Since the authors mention that Kyoto-2006+ is a classic dataset, the authors should give a comprehensive review of related work using Kyoto-2006+, and it would help me to understand how previous works use the dataset and the contribution significance of the proposed dataset in this paper. Hope the authors can provide more information regarding my concerns above.
>
> **R1.** Thanks for bringing this up. Our **goal** was **to build an anomaly detection benchmark to highlight the natural distribution shifts in stream-like data**.
>
> For completeness, **we have considered various alternatives**, not limited to network traffic, but also logs datasets and other multivariate timeseries. The majority of available datasets have a **small timespan**, not being suitable for observing this natural data shift. Even for logs datasets that generally have timespans of hundreds of days, or **analysis did not emphasize convincing distribution shifts**, neither from the perspective of baseline anomaly detection models nor from the perspective of data analysis (t-SNE, Jeffreys divergence, and OTDD analysis). For other datasets, like the popular anomaly detection NSL-KDD dataset, the **performance of current anomaly detection models is high enough** not to justify further research. For a more detailed analysis, please see the response provided to Reviewer dm8f.
>
> Due to its large timespan, **Kyoto-2006+** was **the most representative**, emphasizing the **distribution shift** of network traffic **over 10 years**.
>
> Kyoto-2006+ has a different **ratio between normal data and anomalies** than most anomaly detection datasets, but we have decided **not to alter the original data distribution**. The purpose of our benchmark is to provide the **tools for developing and testing methods under the distribution shift paradigm**. Irrespective of the corruption level considered during training, the goal is to understand if the models we are building are robust enough to cope with natural distribution shifts.
>
> In Appendix A, Fig. 9, we analyze the **behaviour of the BERT-based anomaly detector under different corruption levels** of the training set, which will typically be the case for a real-world scenario. We observe the same performance evolution between IID, NEAR, and FAR splits, suggesting that the impact of distribution shift is noticeable under various levels of corruption. This highlights the relevance of our benchmark, indicating its potential to advance this field of research.
>
> ---
>
> **Q2.** **[BERT code]** The authors propose a BERT-based anomaly detector, a novel approach to detecting tabular data anomalies. It would be better if the author could open-source the code to facilitate future research.
>
> **R2.** Thanks for pointing this out. We have included the **code for training and evaluating the BERT-based anomaly detector in our [AnoShift repository](https://github.com/bit-ml/AnoShift/tree/main/baselines)**.
>
> ---
>
> **Q3.** **[Baselines code]** The authors claim the paper is a benchmark paper, but the GitHub repo only has the code for the IF baseline, and the paper does not have the implementation details and parameters of the baselines required by the NeurIPS benchmark track CFP. I suggest the authors either add the missing information to make the paper a benchmark one or rewrite the paper to a dataset paper.
>
> **R3.** We appreciate your advice and consequently we have included the **code for running all the considered baselines in our [project’s repository](https://github.com/bit-ml/AnoShift/tree/main/baselines)**. More, we significantly enlarge the initial collection of baselines (both with results and their code) with 6 new methods (Tab. 1), while the takeaways and insights stay the same.
>
> ---
> (continues in Part2)

---

> > ### Author Response · Authors · 2022-08-16
> > **Response to Reviewer 7yuQ - Part2**
> >
> >
> > **Q4.** **[Fig. 4 update]** Figure 4 shows the distribution shift intuitively, but it takes me some time to interpret, I suggest the authors explain the differences between rows and columns.
> >
> > **R4.** For a better understanding, we have **updated Fig. 4** to include a legend highlighting that each year has an associated color, used to represent its point clouds. Also, we have **included additional explanations**: “In Fig. 4, we introduce the comparison between pairs of yearly splits and the whole figure can be interpreted as a similarity matrix, each cell (i,j) illustrating the similarity between point clouds of year i vs. year j. Each row illustrates the point cloud of the corresponding year overlapped on all the other point clouds. At the same time, each column presents the point clouds of the corresponding year below all the other point clouds for a better understanding of the distribution shifts. We can observe that points move away as we increase the temporal gap between their corresponding years.”

---

> ### Comment · Reviewer_7yuQ · 2022-08-29
> **Thanks for the update from the authors.**
>
> The authors have updated the code repository with more baselines. Even though the data is not ideal for typical anomaly detection problems, I appreciate the authors' effort in investigating other datasets and making an in-depth analysis of the dataset and baselines. Hope the authors have a maintenance plan for the benchmarking, including adding more suitable datasets and baselines. I will keep my current score.

---

### Official Review · Reviewer_dm8f · 2022-07-26
**A benchmark for unsupervised anomly detection but with limited contribution**

**Rating:** 4
**Confidence:** 5
**Clarity:** This article is well organized and wr…

**Strengths:**

1.It introduces the distribution shift problem to an unsupervised learning setting: Anomaly detection. The visualization and statistical analyses indicate the inherent shift of Network traffic anomaly detection data over time and the experimental results of IID models on shift data show the accuracy decrease caused by distribution shift. The proposed problem is practical and meaningful.

2.This paper is well-written and easy to follow, the details of how to construct shift data are provided. The experimental setup is reasonable and could support the author's claim.

**Weaknesses:**

1. As I know, there exist many time series anomaly detection datasets, such as KDD99[1], SWaT[2], and SMAP[3], but this paper includes only one dataset edited from an existing datasets, which is not enough for a solid benchmark research.

2.More comparison baselines should be considered, including but not limited to reconstructed-based methods, GAN based methods.
3.The finetune mode and distillation mode are not detailed, their pseudocodes should be provided in the Appendix.

[1] Richard Lippmann, Joshua W Haines, David J Fried, Jonathan Korba, and Kumar Das. The 1999 DARPA off-line intrusion detection evaluation. Computer Networks, 34(4):579 – 595, 2000.

[2] Aditya P Mathur and Nils Ole Tippenhauer. SWaT: A water treatment testbed for research and training on ICS security. In International Workshop on Cyber-physical Systems for Smart Water Networks, pages 31–36, 2016.

[3]KyleHundman,ValentinoConstantinou,ChristopherLaporte,IanColwell,andTomSoderstrom.Detecting spacecraft anomalies using lstms and nonparametric dynamic thresholding. In International Conference on Knowledge Discovery & Data Mining, pages 387–395, 2018.

**Additional Feedback:**

1. In Table 1, the performance of Bert on NEAR is better than IID, What causes this phenomenon？ Is it because Bert take sequence as input?

2. Check the coordinate values of the diagram. For example, in Fig 8, ROC_AUC and PR-AUC should be in the range [0,1].

**Correctness:**

The dataset is built on a previous dataset. It utilizes ROC-AUC and PR-AUC to evaluate the baselines, and the experimental design is generally OK. However, the latest methods of anomaly detection were not included, and experiments on just one data set are not convincing enough.

**Documentation:**

Yes, the code is provided for reproducibility.

**Ethics:**

This paper do not involve any ethical concerns.

**Relation To Prior Work:**

Yes. It claims that this is the first work introduce distribution shift to unsupervised learning.

**Summary And Contributions:**

This paper proposes a distribution shift benchmark for unsupervised anomaly detection. The dataset is built over a network traffic data Kyoto-2006+ and the test data is split into three distributions IID, NEAR, and FAR chronologically. Some analyses based on t-SNE visualization, Jeffreys divergence, and Optimal Transport Dataset Distance are performed to show the inherent shift of Kyoto-2006+ over time. Furthermore, it conducts experiments on this dataset with some classic anomaly detection methods and a Bert-based model to check the impact of the distribution shift on IID models. A basic distillation technique is introduced to improve the anomaly detection performance on shift data.

---

> ### Author Response · Authors · 2022-08-16
> **Response to Reviewer dm8f - Part1**
>
> We thank the reviewer for the in-depth feedback and for pointing out some key aspects for improving our benchmark. We took them very seriously, and we hope we properly addressed them
>
> ---
>
> **Q1**. **[More datasets]** As I know, there exist many time series anomaly detection datasets, such as KDD99, SWaT, and SMAP, but this paper includes only one dataset edited from an existing datasets, which is not enough for a solid benchmark research.
>
> **R1**. Thank you for asking this question! We did not show the process of how we chose this dataset, the validations it passed through, and why we consider it to be one of the few relevant in the distribution-shift context for stream-like data. We performed an in-depth analysis over a large number of datasets, even contacting several authors to find out if their dataset is proper for distribution shift (e.g., if there are properties we missed that we can add to it). We wanted it to come from stream-like data (as opposed to the less natural, existing benchmarks on images or text ​​[26, 22, 43, 5, 20]), and we looked for two characteristics that we consider essential for a distribution shift benchmark:
>
> * It spreads over a significant enough time-span such that the distribution shift will naturally occur, rather than being synthetically injected, exhibiting sudden changes
> * It is not solved already (meaning that existing methods do not report almost perfect scores on it)
>
> We have added all these details in Appendix A.3.
>
> ### Network traffic datasets
> We first looked over a wide range of known network traffic datasets for intrusion detection. After analyzing them, we concluded that most are artificially created, with injected samples, in very restricted scenarios. Only Kyoto-2006+ was a proper dataset, extended over a long enough period of time to show a natural distribution shift:
>
> | Dataset  | Number of samples | Time-span | Other details |
> | ------------- |:-------------:| ------------- |:-------------:|
> | [CIC-IDS2017](https://www.unb.ca/cic/datasets/ids-2017.html)   |  3 mil	| 5 days  	| Different attack types are simulated in different days, no natural shift to be observed
> | [CSE-CIC-IDS2018](https://www.unb.ca/cic/datasets/ids-2018.html)   |  4.5 mil	| 17 days  	| Different attack types are simulated in different days, no natural shift to be observed
> | [UNSW-NB15](https://research.unsw.edu.au/projects/unsw-nb15-dataset)   |   2.5 mil   | 2 days  	| too small |
> | [BoT-IoT](https://research.unsw.edu.au/projects/bot-iot-dataset)   |   73 mil   | 4 days  	|too small |
> | [ToN-IoT](https://research.unsw.edu.au/projects/toniot-datasets)	| 22 mil  | 6 days  	|too small |
> |  NSL-KDD ([fixed](https://www.unb.ca/cic/datasets/nsl.html) KDD99)    |   0.15 mil   | 45 days | max reported  **ROC-AUC   99%** |
> | [LANL](https://arxiv.org/abs/1803.04967)   |  1.6 mil	| 58 days  |max reported  **ROC-AUC  99%**|
> | AAD   |   1.8 mil   | 90 days 	| an internally build dataset, given our current (Bitdefender) labeling we get **ROC-AUC over 98%**, making it improper for a new baseline |
> | [Kyoto-2006+](http://www.takakura.com/Kyoto_data/)   |  806M	| 10 years  	| |
>
>
> ### System logs datasets
> We next focused our attention on system logs since the time-span is usually more extensive in these datasets and the natural distribution shift is more probable. But under our analysis (t-SNE, Jeffreys divergence, OTDD, multiple baselines), these datasets did not exhibit a clear distribution shift over time, so we decided to analyze them further until we reach some concludent results. We used Drain and Spell as log parsers, and we report here the results (ROC-AUC) using the LogAnomaly baseline.
>
> | Dataset - Preprocessor  | Number of samples | Time-span | IID | NEAR | FAR|Split proportion|
> | ------------- |:-------------:| ------------- |:-------------:|:-------------:|:-------------:|:-------------:|
> | [HDFS](https://zenodo.org/record/3227177#.YvZIGuxBwpM)  - Drain |   11 mil   | 40h      | 54%| 66%|57% |6-6-6|
> | [BGL](https://www.usenix.org/cfdr-data#hpc4) -	Spell/Drain   |  4.7 mil    | 214 days      | 67%/68%| 43%/73%| 45%/35%|2-3-2|
> | [Thunderbird](https://www.usenix.org/cfdr-data#hpc4)	-  Spell/Drain |    211 mil  | 244 days       | 72%/71%| 72%/72%| 76%/75%|3-3-3|
> | [Liberty](https://www.usenix.org/cfdr-data#hpc4)  | 266 mil     | 315 days      | anomalies only in 4 consecutive months (all grouped)| |
> | [Spirit-CMU](https://www.usenix.org/cfdr-data#hpc4) - Spell     |  272 mil    | 570 days      | 80%|67% |72% |6-5-3|
>
>
> [LogAnomaly] Weibin Meng et al., Loganomaly: Unsupervised detection of sequential and quantitative anomalies in unstructured logs, IJCAI 2019
>
> (continues in Part 2)

---

> > ### Author Response · Authors · 2022-08-16
> > **Response to Reviewer dm8f - Part2**
> >
> > ### Multi-variate timeseries datasets
> > We next looked over general multi-variate timeseries datasets, but the most popular ones are quite small and almost perfectly solved already.
> >
> > | Dataset  | Number of samples | Time-span | max reported unsup ROC-AUC |
> > | ------------- |:-------------:| ------------- |:-------------:|
> > | [SMAP - Soil Moisture Active Passive](https://smap.jpl.nasa.gov/data/)  | 0.5 mil     | 7-14 days|99% |
> > | [SWaT - Secure Water Treatment](https://itrust.sutd.edu.sg/testbeds/secure-water-treatment-swat/)  |   0.9 mil   | 11 days|85% |
> > | [WADI - Water Distribution](https://itrust.sutd.edu.sg/testbeds/water-distribution-wadi/) | 0.96 mil     |16 days   |90% |
> > | [SMD - Server Machine Dataset](https://github.com/NetManAIOps/OmniAnomaly/tree/master/ServerMachineDataset)  |  1.4 mil    |  35 days      |99% |
> > | [MSDS - Multi-Source Distributed System](https://zenodo.org/record/3549604#.YvZLEexBwpM)  |   0.3 mil   |  days - months   |91%  |
> > | [PSM - Pooled Server Metrics](https://github.com/eBay/RANSynCoders/tree/main/data)  |      | 147 days| 98% |
> > | [MSL - Mars Science Laboratory](https://github.com/khundman/telemanom) |  0.13 mil    | - |99% |
> > | [NAB - Numenta Anomaly Benchmark](https://numenta.com/machine-intelligence-technology/numenta-anomaly-benchmark/)  |  0.37 mil    | - |99% |
> > | [MBA - MIT-BIH Supraventricular Arrhythmia](https://physionet.org/content/svdb/1.0.0/)  |  0.2 mil   | 78 half-hour ECGs |99% |
> >
> > ### Datasets exemplified by the reviewer
> >
> > Nevertheless, we split the datasets the reviewer proposed in 4 parts: TRAIN+IID, NEAR, and FAR (chronologically ordered, following the proposed AnoShift setup) and run IsoForest as baseline. It can be easily observed that either the scores are way too high for considering it a challenging baseline (NSL-KDD), or the shift does not appear in the way the datasets are built (on SWaT and SMAP). We did not further investigate this, but one reason why the shift is not observed in SWaT and SMAP might be the short time-span of the datasets (1-2 weeks of data). It means they capture a very local behavior and cannot contain a natural shift. Yet, distribution shift is a known problem in real-world scenarios, on data evolving naturally over a sufficiently large period (as we see in Kyoto-2006+). We hope our work will further encourage the research of this phenomenon and help build more robust models and design future distribution shift datasets.
> >
> >
> > | Dataset  | IID | NEAR | FAR | Split proportion |
> > | ------------- |:-------------:| ------------- |:-------------:|:-------------:|
> > | NSL-KDD ([fixed](https://www.unb.ca/cic/datasets/nsl.html) KDD99)   |  98%|98%|94%|3-1-1|
> > | SWaT   | no anomalies | 49% | 92% | 3-1-1|
> > | SMAP   |71%|	53%| 60%|3-1-1|
> >
> > ---
> >
> > (continues in Part3)

---

> > > ### Author Response · Authors · 2022-08-16
> > > **Response to Reviewer dm8f - Part3**
> > >
> > > **Q2**. **[More Baselines]** More comparison baselines should be considered, including but not limited to reconstructed-based methods, GAN based methods.
> > >
> > > **R2**.  Thank you very much for this suggestion. We have **considered additional baseline models** to strengthen our proposed benchmark. We have chosen **various recent approaches** ranging from probabilistic (ECOD, COPOD) to deep learning models employing either self-supervised contrastive learning (InternalContrastiveLearning), variational autoencoders (VAE), graph neural networks (LUNAR) or generative models (SO-GAAL). We observe **the same performance degradation**, highlighting the inherent distribution shift and the high-performance illusion induced by training and testing the models in the same distribution.  For VAE, LUNAR, and InternalContrastiveLearning,  we observe a continuous performance degradation from IID to NEAR and FAR splits. For ECOD and COPOD, both the performance of NEAR and FAR splits are below random although the IID performance is impressive. An interesting result is obtained for the SO-GAAL method, where we observe a performance increase between IID and NEAR splits, while the performance drastically degrades for the FAR split. This is consistent with the results obtained for our BERT-based anomaly detection model, indicating that these two approaches may be suitable for anomaly detection in the presence of milder distribution shifts (between IID and NEAR splits), learning some train specific biases. Yet, under more severe distribution shifts (between IID and FAR splits), the models fail, proving they are not robust enough. This opens up the possibility of new research directions (as detailed in R4), highlighting the importance and utility of our proposed AnoShift benchmark.
> > >
> > > In the table below we present the ROC-AUC (&uarr;) obtained for the 6 baselines over our IID, NEAR, and FAR splits.
> > >
> > > | | IID | NEAR  | FAR |
> > > |:------------- |:-------------|:-------------|:-------------|
> > > | **ECOD** (2022) | 80.76  | 24.09  | 41.39 |
> > > | **COPOD** (2020) | 83.11   |  30.31 | 42.29 |
> > > | **SO-GAAL** (2019) | 61.82 &pm; 0.28 | 81.24 &pm; 1.19 | 39.74  &pm; 0.21 |
> > > | **VAE** (2014) | 69.98  | 33.53 | 19.21  |
> > > | **LUNAR** (2022)  | 81.19 &pm; 0.72 | 46.10 &pm; 0.66 | 26.57 &pm; 0.17 |
> > > | **InternalContrastiveLearning** (2022) | 84.15 &pm; 1.15 | 51.89 &pm; 2.89 | 22.33 &pm; 0.60|
> > >
> > > [Internal Contrastive Learning] Shenkar T and Wolf L. Anomaly detection for tabular data with internal contrastive learning.  ICLR 2022
> > >
> > > [ECOD] Zheng Li, Yue Zhao, Xiyang Hu, Nicola Botta, Cezar Ionescu, and H. George Chen. Ecod: unsupervised outlier detection using empirical cumulative distribution functions. IEEE Transactions on Knowledge and Data Engineering, 2022.
> > >
> > > [COPOD] Zheng Li, Yue Zhao, Nicola Botta, Cezar Ionescu, and Xiyang Hu. COPOD: copula-based outlier detection.  In IEEE International Conference on Data Mining (ICDM). IEEE, 2020.
> > >
> > > [LUNAR] Adam Goodge, Bryan Hooi, See-Kiong Ng, and Wee Siong Ng. Lunar: unifying local outlier detection methods via graph neural networks. In Proceedings of the AAAI Conference on Artificial Intelligence, 2022.
> > >
> > > [SO-GAAL] Yezheng Liu, Zhe Li, Chong Zhou, Yuanchun Jiang, Jianshan Sun, Meng Wang, and Xiangnan He. Generative adversarial active learning for unsupervised outlier detection. IEEE Transactions on Knowledge and Data Engineering, 2019.
> > >
> > > [VAE] Diederik P Kingma and Max Welling. Auto-encoding variational Bayes. ICLR 2014
> > >
> > > ---
> > >
> > > **Q3**. **[Finetune and Distillation]** The finetune mode and distillation mode are not detailed, their pseudocodes should be provided in the Appendix.
> > >
> > > **R3**. Thank you for pointing out our lacklustre explanation of the different training strategies from our experiments. We addressed this by adding a detailed pseudocode in Appendix A.2 and by including the code to reproduce our experiments, as well as a notebook tutorial that compares the training strategies on a data subset, in our [GitHub repository](https://github.com/bit-ml/AnoShift/blob/main/baselines/iid_finetune_distill_comparison.ipynb).
> > >
> > > ---
> > >
> > > (continues in Part4)

---

> > > > ### Author Response · Authors · 2022-08-16
> > > > **Response to Reviewer dm8f - Part4**
> > > >
> > > > **Q4**. **[BERT - NEAR vs IID]** In Table 1, the performance of Bert on NEAR is better than IID, What causes this phenomenon? Is it because Bert take sequence as input?
> > > >
> > > > **R4**. We thank the reviewer for pointing this out. We further present 3 different points of view for addressing it. We added this discussion in the main paper and clarify in Tab. 1 caption:
> > > >
> > > > a. For better understanding this result, we further tested **more baselines** (both classical and new methods - Tab. 1). We observe that **in general, the scores are decreasing from IID to NEAR**, for almost all tested methods (9 out of 11, best viewed in Fig. 11 - Appendix). But BERT and the SO-GAAL are exceptions. We argue that this happened due to certain method particularities, rather than the dataset (also motivated by our t-SNE, Jeffreys divergence, and OTDD analysis). Moreover, the ROC-AUC evolution for the two methods is explained by a very large PR-AUC-outliers compared with others, while the PR-AUC-inliers follows the gradual degradation over time (Tab. 2 in Appendix). To emphasize the differences, we also add the best score per split (different from BERT or SO-GAAL).
> > > >
> > > > | ROC-AUC | Method  | IID| NEAR| FAR|
> > > > | ------------- |:-------------| ------------- |:-------------:|:-------------:|
> > > > || **BERT** | 84.54%| 86.05% | 28.15%|
> > > > || **SO-GAAL**| 61.82%| 81.24% |39.74%|
> > > > || **(different) Best per split** | 86.58 %| 74.45% | 46.96 %|
> > > >
> > > >
> > > >
> > > > | PR-AUC-in | Method  | IID| NEAR| FAR|
> > > > | ------------- |:-------------| ------------- |:-------------:|:-------------:|
> > > > || **BERT** |74.61%|58.94% |8.22%|
> > > > || **SO-GAAL** |61.53%|52.66% |9.97%|
> > > > || **(different) Best per split** |78.77%|44.71% |13.14%|
> > > >
> > > >
> > > >
> > > > | PR-AUC-out | Method  | IID| NEAR| FAR|
> > > > | ------------- |:-------------| ------------- |:-------------:|:-------------:|
> > > > || **BERT** |89.83%|95.96% |78.38%|
> > > > || **SO-GAAL** |65.53%|93.41% |88.59%|
> > > > || **(different) Best per split** |88.81%| 89.60% |86.80%|
> > > >
> > > >
> > > >
> > > > b. We have an additional confirmation for the distribution shift in AnoShift setting, by testing several **supervised methods**, and we see there that **the performance also degrades over time** (see Tab. 5 and Fig. 12 in the Appendix).
> > > >
> > > > c. We also present **some insights** which we consider that might **prevent BERT** (or Masked Language Models in general) **to be a robust anomaly detector**, and we think that might be interesting for future research:
> > > > * **MLM as anomaly detector**: Even with a large number of parameters, the anomaly performance of BERT is based on the perplexity score when predicting several masked features in the sample. So if the features are not correlated, the MLM model might be unable to learn something useful, which might result in learning some specific training set biases, failing to generalize on further data (eg. lower score on FAR wrt other baselines).
> > > > * **MLM with the training vocabulary**: If the model only learns based on the training data, all the new words (in our case, features from a new sample) will be considered UNK, and the score might get artificially inflated (since from a point further it is easier to predict UNK instead of the right word). This might be a disadvantage of vocabulary based methods as opposed to other classic approaches.
> > > >
> > > > [SO-GAAL] Yezheng Liu, Zhe Li, Chong Zhou, Yuanchun Jiang, Jianshan Sun, Meng Wang, and Xiangnan He. Generative adversarial active learning for unsupervised outlier detection. IEEE Transactions on Knowledge and Data Engineering, 2019.
> > > >
> > > > ---
> > > >
> > > > **Q5**. **[Fig. 8 update]** Check the coordinate values of the diagram. For example, in Fig 8, ROC_AUC and PR-AUC should be in the range [0,1].
> > > >
> > > > **R5**. Thank you for pointing out that Fig. 8 needs additional explanation. We have illustrated the performance of **_finetune_** and **_distill_** strategies as differences towards the performance of the **_iid_** strategy. For the whole paper, we have reported ROC-AUC and PR-AUC as percentages, thus in the range [0,100]. Consequently, **the values should not be restricted to the range [0,1] as they highlight differences between values in the range [0,100]**. Moreover, we can even have negative values if one of the considered strategies performs worse than **_iid_**.
> > > >
> > > > We have **updated Fig. 8** to better indicate the meaning of each axis.
> > > >
> > > > ---

---

### Official Review · Reviewer_aNvW · 2022-07-27
**An unsupervised temporal anomaly dataset**

**Rating:** 7
**Confidence:** 3
**Clarity:** Yes, the paper is well written.

**Strengths:**

- This is quite a useful datasets especially because data is collected over a long time period.
- The tasks are well-defined and studies.
- Data shift is an upcoming research area in ML, and the dataset would be very useful.

**Weaknesses:**

The code repository is under developed. Tutorials are not well-structured, they do not even have comments.

**Additional Feedback:**

- the websites must be better designed, hosting plans should be made clear.
- Github repo should have better tutorials or starting points

**Correctness:**

- The dataset is constructed in a sound way, and the evaluation methods (e.g., Pr-AUC) are appropriate.

**Documentation:**

Among these, the availability and maintenance, and ethical and responsible use are lacking. As the data resides on Takakura.com, guarantees about hosting and plans must have been included.

**Ethics:**

I do not see any issues with ethics.

**Relation To Prior Work:**

Yes, the prior work is discussed in the related work section.

**Summary And Contributions:**

The authors provide a long-time data created by using honeypots to detect anomalous traffic. The characteristics of the data have been studied by using TSNE and optimal transport. The data is divided into train, near and far sets to understand the performance of anomaly detection methods.

---

> ### Author Response · Authors · 2022-08-16
> **Response to Reviewer aNvW**
>
> Thank you for your overall positive feedback and for pointing out ways to improve our project’s repository. We are confident that this will have a significant impact on our work.
>
> Below we address each of your comments:
>
> ---
>
> **Q1**. **[AnoShift repository]** The code repository is under developed. Tutorials are not well-structured, they do not even have comments. Github repo should have better tutorials or starting points
>
> **R1**. We have significantly **updated the project’s repository** (https://github.com/bit-ml/AnoShift) to include:
>
> - **Tutorials** for using AnoShift
> - **Implementation** of our proposed **BERT-based anomaly detector** (including pretrained models and Pre-saved Word Level tokenizer)
> - **Implementation** for all considered **baselines**
> - Full **preprocessing code**
>
> ---
>
> **Q2**. **[Availability update]** Among these, the availability and maintenance, and ethical and responsible use are lacking. As the data resides on Takakura.com, guarantees about hosting and plans must have been included. The websites must be better designed, hosting plans should be made clear.
>
> **R2**. The **preprocessed Kyoto-2006+** is also **available for download from our own storage**: https://share.bitdefender.com/s/9D4bBE7H8XTdYDB and we are responsible for keeping this data available.
>
> Thank you for bringing up the discussion about the **availability of raw data**.  We are **not able to guarantee** its availability, as it **depends on Takakura.com**. To avoid further problems, we have published our preprocessed version We updated this detail in (Appendix B.3).
>
> To the best of our knowledge there is **no sensitive data leaked in the preprocessed dataset** and we consider that there are no ethical concerns regarding this dataset (Appendix B.2).

---

### Author Response · Authors · 2022-08-16
**Summary of Our Responses**

We sincerely thank all the reviewers for their insightful comments, questions, and suggestions, which inspired us to better emphasize our work's value. We carefully address each review and provide individual responses.

To make it easier to follow for the reviewers, we uploaded the revised main content and supplementary material in one single file as **supplementary material**.

The primary changes are summarized below:

* **More baselines**: We provide results and evaluation code for other 6 competitive baseline methods; these new experiments support all our key insights and conclusions.

* **Analysis of multiple datasets**: We have detailed the analysis we have performed over multiple types of datasets before choosing Kyoto-2006+ as being one of the few relevant for the distribution-shift problem.

* **Improved Github repo**: We significantly improved our github repository, with detailed code for all our baselines.

* **Supervised vs Unsupervised**: We added 6 supervised baselines to emphasize on the gap between the supervised and unsupervised methods.

* **BERT - NEAR vs IID** discussions: We provide additional analysis of our experimental results.

We hope that the detailed answers and the additional results provided to the reviewers will convince them that our benchmark is relevant for the field and can be helpful to move the distribution shift works further.

---

### Meta-Review · Area_Chair_eG2y · 2022-09-09

**Recommendation:** Accept
**Confidence:** 4

**Metareview:**

Four out of five reviewers recommend acceptance of the paper with a score of 7. These reviewers praise the quality and importance of the dataset. Moreover, all reviewers acknowledge the quality of the writing.

The authors have constructively engaged into the discussion with the reviewers, and have put a lot of effort into drastically improving their submission based on all reviewers' comments. The weaknesses raised by the reviewers have been successfully addressed.

In particular, the reviewer (dm8f) who gave a score of 4 raised the following concerns:

1. There exist other time series anomaly detection datasets.
2. More anomaly detection methods should be considered in the experiments.
3. The finetune mode and distillation mode are not detailed, their pseudocodes should be provided.

The authors have addressed these concerns as follows:

1. In their answer, the authors went through an extensive list of existing benchmarks and justified why they are not appropriate to solve their focused problem, and how their dataset differ from those. The main added value of the proposed dataset is its timespan and the fact that its occurring distribution shifts are natural (as opposed to synthesized).
2. The authors have expanded their experiments with more anomaly detection models. The new results corroborate the previous conclusions.
3. The authors have added the pseudocode algorithm in Appendix and have added code to replicate their experiments in the git repository.

I believe that the authors have sufficiently addressed the reviewer's concerns (and more).

For these reasons, I recommend acceptance of the paper.

---

### Decision · Program_Chairs · 2022-09-16

Accept